# MODEL-BASED REINFORCEMENT LEARNING FOR PARAMETERIZED ACTION SPACES

## ABSTRACT

We propose a novel model-based reinforcement learning algorithm—Dynamics Learning and predictive control with Parameterized Actions (DLPA)—for Parameterized Action Markov Decision Processes (PAMDPs). The agent learns a parameterized-action-conditioned dynamics model and plans with a modified Model Predictive Path Integral control. We theoretically quantify the difference between the generated trajectory and the optimal trajectory during planning in terms of the value they achieved through the lens of Lipschitz Continuity. Our empirical results on several standard benchmarks show that our algorithm achieves superior sample efficiency and asymptotic performance than state-of-the-art PAMDP methods.

## 1 INTRODUCTION

Reinforcement learning (RL) has gained significant traction in recent years due to its proven capabilities in solving a wide range of decision-making problems across various domains, from game playing [27] to robot control [37; 24]. One of the complexities inherent to some RL problems is a discrete-continuous hybrid action space. For instance, in a robot soccer game, at each time step the agent must choose a discrete action (move, dribble or shoot) as well as the continuous parameters corresponding to that chosen discrete action, e.g. the location (x,y) to move/dribble to. In this setting, Parameterized Action Markov Decision Processes (PAMDPs) [26], each discrete action is parameterized by some continuous parameters, and the agent must choose them together at every timestep. PAMDPs model many real world scenarios like RTS Games [42] or Robot Soccer [19].

Compared to just discrete or continuous action space, Reinforcement Learning with Parameterized action space is able to let the agent perform more structural exploration and solve more complex tasks with the semantically more meaningful action space [19]. Recent papers provide various approaches for RL in the PAMDP setting [3; 8; 7; 23]. However, to the best of our knowledge, all previous methods are model-free. By contrast, in continuous/discrete-only action spaces, deep model-based reinforcement learning has shown better performance than model-free approaches in many complex domains [14; 15; 16; 17], in terms of both sample efficiency and asymptotic performance We therefore seek to leverage the high sample efficiency of model-based RL in PAMDPs.

We propose a model-based RL framework tailored for PAMDPs—Dynamics Learning and predictive control with Parameterized Actions (DLPA). Our approach first learns a parameterized-action-conditioned dynamics model. It then performs Model Predictive Control [34] by modifying the Cross-Entropy Method (CEM) to harness the inherent structure of parameterized action spaces while mitigating their complexities. We also provide theoretical analysis regarding DLPA's performance guarantee and sample complexity. Our empirical results on 8 different PAMDP benchmarks show that DLPA achieves better or comparable asymptotic performance with significantly better sample efficiency than all the state-of-the-art PAMDP algorithms. We also find that DLPA succeeds in tasks with extremely large parameterized action spaces where prior methods cannot succeed without learning a complex action embedding space, and converges much faster. The proposed method is even better than the method that has a customized action space compression algorithm as the original parameterized action space becomes larger. To the best of our knowledge, our work is the first method that successfully applies model-based RL to PAMDPs.

## 2 BACKGROUND

### 2.1 PARAMETERIZED ACTION MARKOV DECISION PROCESSES

Markov Decision Processes (MDPs) form the foundational framework for many reinforcement learning problems, where an agent interacts with an environment to maximize some cumulative reward. Traditional MDPs are characterized by a tuple $\{S, A, T, R, \gamma\}$, where $S$ is a set of states, $A$ is a set of actions, $T$ denotes the state transition probability function, $R$ denotes the reward function, and $\gamma$ denotes the discount factor. Parameterized Action Markov Decision Processes (PAMDPs) [26] extend the traditional MDP framework by introducing the parameterized actions, denoted as a tuple $\{S, M, T, R, \gamma\}$, where $M$, differs from traditional MDPs, is the parameterized action space that can be defined as $M = \{(k, z_k)|z_k \in Z_k$ for all $k \in \{1, \cdots, K\}\}$, where each discrete action $k$ is parameterized by a continuous parameter $z_k$, and $Z_k$ is the space of continuous parameter for discrete action $k$ and $K$ is the total number of different discrete actions. Thus, we have the dynamic transition function $T(s'|s, k, z_k)$ and the reward function $R(r|s, k, z_k)$.

### 2.2 MODEL PREDICTIVE CONTROL

In Deep Reinforcement Learning, Model-free methods usually learn a policy parameterized by neural networks that learns to maximize the cumulative returns $\mathbb{E}_\tau[\sum_t \gamma^t R(s, a)]$. In the model-based domain, since we assume we have access to the learned dynamics model, we can use Model Predictive Control [10] to plan and select the action at every time step instead of explicitly learn to approximate a policy function. Specifically, at time step $t$, the agent will first sample a set of action sequences with the length of horizon $H$ for states $s_t : s_{t+H}$. Then it will use the learned dynamics model to take in actions as well as the initial states and get the predicted reward for each time step. Then the cumulative reward for each action sequence will be computed and the action sequence with the highest estimated return will be selected to execute in the real environment. Cross-Entropy Method (CEM) [35] is often used together with this planning procedure, which works by iteratively fitting the parameters of the sampling distributions over the actions.

## 3 RELATED WORK

Several model-free RL methods have been proposed in the context of deep reinforcement learning for PAMDPs. PADDPG [19] builds upon DDPG [24] by letting the actor output a concatenation of the discrete action and the continuous parameters for each discrete action. Similar ideas are proposed in HPPO [7], which is based on the framework of PPO [38]. P-DQN [42; 3] is another framework based on the actor-critic structure that maintains a policy network that outputs continuous parameters for each discrete action. This structure has the problem of computational efficiency since it computes continuous parameters for each discrete action at every timestep. Hybrid MPO [29] is based on MPO [1] and it considers a special case where the discrete part and the continuous part of the action space are independent, while in this paper we assume the two parts have strong correlations. HyAR [23] proposes to construct a latent embedding space to model the dependency between discrete actions and continuous parameters, achieving the best empirical performance among these methods. However, introducing another latent embedding space can be computationally expensive and the error in compressing the original actions may be significant in complex tasks. While all these methods have been shown to be effective in some PAMDP problems, to the best of our knowledge, no work has successfully applied model-based RL to PAMDPs, even though model-based approaches have achieved high performance and excellent sample efficiency in MDPs.

Most of existing (Deep) Model-based Reinforcement Learning methods can be classified into two categories in terms of how the learned model is used. The first category of methods learns the dynamics model and plans for credit assignment [6; 44; 20; 13; 25; 21; 4; 43; 36; 30]. A large number of algorithms in this category involves planning with random shooting [28] or Cross-Entropy Method [35; 5]. The other way is to use the learned model to generate more data and explicitly train a policy based on that [32; 12; 14; 39; 16], which is also known as Dyna [40]-based methods. There are also algorithms combining model-free and model-based methods [17]. But none of these methods are in the parameterized action (PAMDP) settings.

# 4 DYNAMICS LEARNING AND PREDICTIVE CONTROL WITH PARAMETERIZED ACTIONS

We propose DLPA, a framework that learns a dynamic transition model for parameterized action space and further use the model to plan and interact with the environment. Our algorithm is summarized in Figure 1 and Algorithm 1.

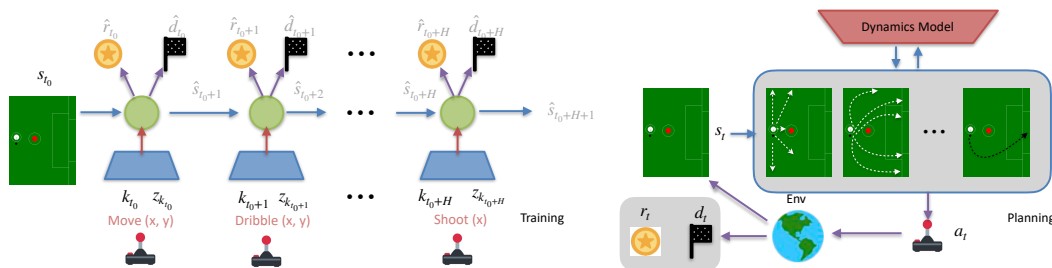

Figure 1: Dynamics Learning and predictive control with parameterized actions (DLPA). Left: Inference of dynamics during training. **Variables colored with default black are those we feed as input to the dynamics model. Variables colored with grey are those generated from the dynamics model.** Right: Planning and interacting with the environment. At each time step we execute only the first action from the sampled trajectory. White lines are example rollout trajectories from DLPA.The black line denotes the final selected rollout trajectory for one planning step.

## 4.1 DYNAMICS MODEL WITH PARAMETERIZED ACTIONS

To perform Model Predictive Control with Parameterized Actions, DLPA requires learning the following list of model components:

$$\text{Transition predictor: } \hat{s}_{t+1} \sim T_\phi(\hat{s}_{t+1}|s_t, k_t, z_{k_t}),$$

$$\text{Continue predictor: } \hat{c}_t \sim p_\phi(\hat{c}_t|s_{t+1}),$$

$$\text{Reward predictor: } \hat{r}_t \sim R_\phi(\hat{r}_t|s_t, k_t, z_{k_t}),$$

where we use $c_t$ to denote the episode continuation flag. Given a state $s_t$ observed at time step $t$, a discrete action $k_t$ and the corresponding continuous parameter $z_{k_t}$, the transition predictor and the reward predictor $T_\phi$ and $R_\phi$ predict the next state $\hat{s}_{t+1}$ and reward $\hat{r}_{t+1}$ respectively. The Continue predictor outputs a prediction $\hat{c}_t$ for whether the trajectory continues at time step $t+1$ given the state $s_{t+1}$.

All the above components are implemented in the form of neural networks and we use $\phi$ to denote the combined network parameters. Specifically, the first three components are implemented with networks with stochastic outputs. i.e., we model the output distribution as a Gaussian and the outputs of the networks are mean and standard deviation of the distribution. We leverage reparameterization trick [22] to allow computing gradients for the sampled $\hat{s}, \hat{r}, \hat{c}$'s.

We train these components through minimizing the loss below:

$$\mathcal{L}_{joint} = \mathbb{E}_{\{s_t, k_t, z_{k_t}, r_t, s_{t+1}, c_t\}_{t_0:t_0+H}} \sum_{t_0}^{t_0+H} \beta^{t-t_0} \Big\{ \lambda_1 \|T_\phi(\hat{s}_{t+1}|\hat{s}_t, k_t, z_{k_t}) - s_{t+1}\|_2^2$$

$$+ \lambda_2 \|R_\phi(\hat{r}_{t+1}|\hat{s}_t, k_t, z_{k_t}) - r_{t+1}\|_2^2 + \lambda_3 \|p_\phi(\hat{c}_t|\hat{s}_{t+1}) - c_t\|_2^2 \Big\}, \tag{1}$$

where $\hat{s}_{t_0} = s_{t_0}$. We use $H$ to denote the planning horizon and $t_0$ to denote the start time step. $\beta$ denotes the hyperparameter we use to control the weight of the loss term. The weight of the sum of the loss will be lower if $t$ is closer to the end time step $t_0 + H$. $\lambda$ denotes the weight of each prediction loss term.

Specifically, at each training step, we first sample a batch of trajectories $\{s_t, k_t, z_{k_t}, r_t, s_{t+1}, c_t\}_{t_0:t_0+H}$ from a replay buffer. Then we do the inference procedures as shown in Figure 1 Left. We give the dynamics model the sampled $s_{t_0}, k_{t_0}, z_{k_{t_0}}$ at first, and we get the predictions of next state $\hat{s}_{t_0+1}$, reward $\hat{r}_{t_0}$ and continuation flag $\hat{c}_{t_0}$ for the first time step. Then, we iteratively let our dynamics model predict the transitions with the sampled parameterized actions and **the predicted state from last time step**. At the end we take the weighted sum of all the prediction losses for state, reward and termination and update the model as described in Equation 1. And the gradients from the loss term for the last time step $t_0 + H$ will be backpropagated all the way to the first inference at time step $t_0$. That is to say, what we give our model as input during training is $\{s_{t_0}, k_{t_0}, z_{k_{t_0}}, k_{t_0+1}, z_{k_{t_0+1}}, \cdots, k_{t_0+H}, z_{k_{t_0+H}}\}$ which contains only the start state $s_{t_0}$ without the other ground truth states in the trajectory. And we use this information to infer all the other information contained in the sampled trajectory and calculate the loss. In contrast, most prior model-based RL algorithms compute the prediction loss for each individual transition by giving the model the ground truth state as input at every time step (functions of ground-truth states v.s. fucntions of intermediate predictions). Intuitively, as we will plan into the future with the exact same length $H$ during planning, our choice allows gradients to flow back through time and assign credit more effectively than the alternative. That is to say, it will help the agent focus on H-step prediction for the cumulative return, which is more important for the planning afterwards. As we show in the experiment section, we empirically find out that, by learning to infer several steps into the future instead of just the next step, the downstream planning tends to get better performance and thus benefits the whole training process.

---

**Algorithm 1** DLPA

---

**Require:** Initialize Dynamics models $T_\phi(\hat{s}_{t+1}|s_t, k_t, z_{k_t}), R_\phi(\hat{r}_t|s_t, k_t, z_{k_t}), p_\phi(\hat{c}_t|s_{t+1})$, planning
    horizon $H$, a set of parameters $\mathcal{C}^0$
   **for** Time t = 0 to TaskHorizon **do**
      **for** Iteration $j$=1 to $E$ **do**
         Sample $N$ action sequences with horizon $H$ from $\mathcal{C}^j$
         Forward the dynamics model $T_\phi(\hat{s}_{t+1}|s_t, k_t, z_{k_t})$ to time step $t + H$ with input $s_t$ and the
    sampled action sequences and get $N$ trajectories $\{\tau_i\}_{1:N}$
         Compute the cumulative return for each trajectory $\mathcal{J}_\tau$ with $R_\phi(\hat{r}_t|s_t, k_t, z_{k_t}), p_\phi(\hat{c}_t|s_{t+1})$
         Select the trajectories with top-$n$ cumulative returns
         Update $C^j$ with Equation 5, 6, 7
      **end for**
      Execute the first action, $\{\hat{k}_{t_0}, \hat{z}_{\hat{k}_{t_0}}\}$, in the sampled optimal trajectory
      Receive transitions from the environment and add to replay buffer $\mathcal{B}$
      Sample trajectories $\{s_t, k_t, z_{k_t}, r_t, s_{t+1}, c_t\}_{t_0:t_0+H}$ with from the replay buffer $\mathcal{B}$
      Initialize $\mathcal{L}_{joint}$=0
      **for** $t = t_0 : t_0 + H$ **do**
         $\hat{s}_{t+1} \sim T_\phi(\hat{s}_{t+1}|\hat{s}_t, k_t, z_{k_t})$
         $\hat{c}_t \sim p_\phi(\hat{c}_t|\hat{s}_{t+1})$
         $\hat{r}_t \sim R_\phi(\hat{r}_t|\hat{s}_t, k_t, z_{k_t})$
         $\mathcal{L}_{joint} \leftarrow \mathcal{L}_{joint} + \beta[\lambda_1\|T_\phi(\hat{s}_{t+1}|\hat{s}_t, k_t, z_{k_t}) - s_{t+1}\|_2^2 + \lambda_2\|R_\phi(\hat{r}_{t+1}|\hat{s}_t, k_t, z_{k_t}) -$
    $r_{t+1}\|_2^2 + \lambda_3\|p_\phi(\hat{c}_t|\hat{s}_{t+1}) - c_t\|_2^2]$
      **end for**
      $\phi \leftarrow \phi - \frac{1}{H}\eta\nabla_\phi L_{joint}$
   **end for**

---

## 4.2 Model Predictive Control with Parameterized Actions

Now we introduce the planning part for the proposed algorithm. The planning algorithm is based on Model Predictive Path Integral [41], adapted for the PAMDP setting.

We model the discrete action $k$ follows a multinomial distribution:

$$k \sim Mult(\theta_1^0, \theta_2^0, \cdots, \theta_K^0), \sum_{k=1}^K \theta_k^0 = 1, \theta_k^0 \geq 0, \qquad (2)$$

where the probability for choosing each discrete action $k$ is given by $\theta_k^0$. For the continuous parameter $z_k$ corresponding to each discrete action $k$, we model it as a multivariate Gaussian:

$$z_k \sim \mathcal{N}(\mu_k^0, (\sigma_k^0)^2 I), \ \mu_k^0, \sigma_k^0 \sim \mathbb{R}^{|z_k|}. \tag{3}$$

At the beginning of planning, we initialize a set of independent parameters $\mathcal{C}^0 = \{\theta_1^0, \theta_2^0, \cdots, \theta_K^0, \mu_1^0, \sigma_1^0, \mu_2^0, \sigma_2^0, \cdots, \mu_K^0, \sigma_K^0\}_{t:t+H}$ for each discrete action and continuous parameter over a horizon with length $H$. Recall that $K$ is the total number of discrete actions. Note that next we will update these distribution parameters for $E$ iterations, so for each iteration $j$, we will have $\mathcal{C}^j = \{\theta_1^j, \theta_2^j, \cdots, \theta_K^j, \mu_1^j, \sigma_1^j, \mu_2^j, \sigma_2^j, \cdots, \mu_K^j, \sigma_K^j\}_{t_0:t_0+H}$.

Then, for each iteration $j$, we independently sample $N$ trajectories by independently sampling from the action distributions at every time step $t$ and forwarding to get the trajectory with length $H$ using the dynamics models $T_\phi, R_\phi, p_\phi$ as introduced in the last section. For a sampled trajectory $\tau = \{s_{t_0}, \hat{k}_{t_0}, \hat{z}_{k_{t_0}}, \hat{s}_{t_0+1}, \hat{r}_{t_0}, \hat{c}_{t_0}, \cdots, \hat{s}_{t_0+H}, \hat{k}_{t_0+H}, \hat{z}_{k_{t_0+H}}, \hat{s}_{t_0+1+H}, \hat{r}_{t_0+H}, \hat{c}_{t_0+H}\}$, we can calculate the cumulative return $\mathcal{J}_\tau$ with:

$$\mathcal{J}_\tau = \mathbb{E}_\tau\big[\sum_{t=t_0}^{t_0+H} \gamma^t c_t R_\phi(\hat{s}_t, \hat{k}_t, \hat{z}_{k_t})\big], \text{where } \hat{s}_{t_0} = s_{t_0}. \tag{4}$$

Let $\Gamma_{k,\tau} = \{\hat{k}_t\}_{t=t_0:t_0+H}$ denote the discrete action sequences and $\Gamma_{z,\tau} = \{\hat{z}_{k_t}\}_{t=t_0:t_0+H}$ denote the continuous parameter sequences within each trajectory $\tau$. Then based on the cumulative return of each trajectory, we select the trajectories with top-$n$ cumulative returns and update the set of distribution parameters $\mathcal{C}^j$ via:

$$\theta_k^j = (1-\alpha)\frac{\sum_{i=1}^n e^{\xi \mathcal{J}_{\tau_i}} \Gamma_{k,\tau_i}}{\sum_{i=1}^n e^{\xi \mathcal{J}_{\tau_i}}} + \alpha\theta_k^{j-1}, \tag{5}$$

$$\mu_k^j = (1-\alpha)\frac{\sum_{i=1}^n e^{\xi \mathcal{J}_{\tau_i}} \Gamma_{z,\tau_i} \mathbb{1}\{\Gamma_{k,\tau_i} == k\}}{\sum_{i=1}^n e^{\xi \mathcal{J}_{\tau_i}}} + \alpha\mu_k^{j-1}, \tag{6}$$

$$\sigma_k^j = (1-\alpha)\sqrt{\frac{\sum_{i=1}^n e^{\xi \mathcal{J}_{\tau_i}} (\Gamma_{z,\tau_i} - \mu_k^j)^2 \mathbb{1}\{\Gamma_{k,\tau_i} == k\}}{\sum_{i=1}^n e^{\xi \mathcal{J}_{\tau_i}}}} + \alpha\sigma_k^{j-1}. \tag{7}$$

We use $\xi$ to denote the temperature that controls the trajectory-return weight. Intuitively, for each iteration $j$, we update our sampling distribution over the discrete and continuous actions weighted by the expected return as the returns come from these actions by forwarding our learned dynamics model. The discrete actions and continuous parameters that achieve higher cumulative return will be more likely to be chosen again during the next iteration. For updating the continuous parameters, we add a indicator function as we only want to update the corresponding distributions for those continuous parameters corresponding to the selected $k$. The updated distribution parameters at each iteration is calculated in the form of a weighted sum of the new value derived from the returns and the old value used at the last time step. We use a hyperparameter $\alpha$ to control the weights.

Note that one major modification we make in the MPPI planning process is that we keep a separate distribution over the continuous parameters for each discrete action and update them at each iteration instead of keeping just one independent distribution for the discrete actions and one independent distribution for the continuous parameters. In other words, we let the distribution of the continuous parameters condition on the chosen discrete action during the sampling process. This is an important change to make when using MPPI for PAMDPs as we don't want to throw away the established dependency between the continuous parameter and corresponding the discrete action in PAMDPs.

After $E$ iterations of updating the distribution parameters, we sample a final trajectory from the updated distribution and execute only the first parameterized action, which is also known as receding-horizon MPC similar to previous work [17]. Then we move on to the next time step and do all the planning procedures again. Note that, for the initialization distribution over the parameters, we just copy the one we get at the last time step as a "warm start".

The overall algorithm is described in Algorithm 1. At each environment time step, the agent executes $E$ steps of forward planning while updating the distribution parameters over discrete actions and continuous parameters. Then it uses the first action sampled from the final updated distribution to interact with the environment and add the new transitions into the replay buffer $\mathcal{B}$. Then if it is in training phase, the agent samples a batch of trajectories from the replay buffer, using the steps introduced in Section 4.1 to compute the loss and update the dynamics models.

## 5 ANALYSIS

In this section, we provide some theoretical performance guarantees for DLPA. We quantify the estimation error between the cumulative return of the trajectory generated from DLPA and the optimal trajectory through the lens of Lipschitz continuity. All the proofs can be found in the appendix.

**Definition 5.1.** A PAMDP is $(L_R^S, L_R^K, L_R^Z, L_T^S, L_T^K, L_T^Z)$-Lipschitz continuous if, for all $s \in S$, $k \in \{1, \cdots, K\}$, and $z \in Z$ where $z \sim \omega(\cdot|k)$[1]:

$$|R(s_1, k, \omega)) - R(s_2, k, \omega))| \leq L_R^S d_S(s_1, s_2), W(T(\cdot|s_1, k, \omega)), T(\cdot|s_2, k, \omega))) \leq L_T^S d_S(s_1, s_2)$$

$$|R(s, k_1, \omega)) - R(s, k_2, \omega))| \leq L_R^K d_K(k_1, k_2), W(T(\cdot|s, k_1, \omega)), T(\cdot|s, k_2, \omega))) \leq L_T^K d_K(k_1, k_2)$$

$$|R(s, k, \omega_1) - R(s, k, \omega_2)| \leq L_R^Z d_Z(\omega_1, \omega_2), \ W(T(\cdot|s, k, \omega_1), T(\cdot|s, k, \omega_2)) \leq L_T^Z d_Z(\omega_1, \omega_2)$$

,where $W$ denotes the Wasserstein Metric and $\omega$ denotes the distribution over the continuous parameters given a discrete action type $k$. $d_S, d_K, d_Z$ are the distance metrics defined on space $S, K, Z$. Note that as $k_1, k_2, \cdots$ are discrete variables, we use Kronecker delta function as the distance metric: $d_K(k_j, k_i) = 1, \forall i \neq j$. The definition can be seen as a simple extension of the Lipschitz continuity assumption for regular MDP [33; 31; 2; 11] to PAMDP. Furthermore, in this paper, we follow the Lipschitz model class assumption [2].

Assuming the error of the learned transition model $\hat{T}$ and reward model $\hat{R}$ are bounded by $\epsilon_T$ and $\epsilon_R$ respectively: $W(T(s, k, \omega), \hat{T}(s, k, \omega)) \leq \epsilon_T, |R(s, k, \omega) - \hat{R}(s, k, \omega)| \leq \epsilon_R$, for all $s, k, \omega$. We can derive the following theorem:

**Theorem 5.2.** For a $(L_R^S, L_R^K, L_R^Z, L_T^S, L_T^K, L_T^Z)$-Lipschitz PAMDP and the learned DLPA $\epsilon_T$-accurate transition model $\hat{T}$ and $\epsilon_R$-accurate reward model $\hat{T}$, let $L_{\hat{T}}^S = \min\{L_T^S, L_{\hat{T}}^S\}$, $L_{\hat{T}}^K = \min\{L_T^K, L_{\hat{T}}^K\}$, $L_{\hat{T}}^Z = \min\{L_T^Z, L_{\hat{T}}^Z\}$ and similarly define $L_{\hat{R}}^S, L_{\hat{R}}^K, L_{\hat{R}}^Z$. If $L_{\hat{T}}^S < 1$, then the regret of the rollout trajectory $\hat{\tau} = \{\hat{s_1}, \hat{k_1}, \hat{\omega_1}(\cdot|\hat{k_1}), \hat{s_2}, \hat{k_2}, \hat{\omega_2}(\cdot|\hat{k_2}), \cdots\}$ from DLPA is bounded by:

$$
\begin{aligned}
|\mathcal{J}_{\tau^*} - \mathcal{J}_{\hat{\tau}}| &:= \sum_{t=1}^{H} \gamma^{t-1} |R(s_t, k_t, \omega_t(\cdot|k_t)) - \hat{R}(\hat{s_t}, \hat{k_t}, \hat{\omega_t}(\cdot|\hat{k_t}))| \\
&\leq \mathcal{O}\Big((L_R^K + L_R^S L_T^K)m + H(\epsilon_R + L_R^S \epsilon_T + (L_R^Z + L_R^S L_T^Z)(\frac{m}{H}\Delta_{k,\hat{k}} + \Delta_{\omega,\hat{\omega}}))\Big),
\end{aligned}
\tag{8}
$$

where $m = \sum_{t=1}^{H} \mathbb{1}(k_t \neq \hat{k_t}), \Delta_{\omega,\hat{\omega}} = W(\omega(\cdot|k), \hat{\omega}(\cdot|k)), \Delta_{k,\hat{k}} = W(\omega(\cdot|k), \omega(\cdot|\hat{k})), N$ denotes the number of samples and $H$ is the planning horizon.

Theorem 5.2 quantifies how the following categories of estimation error will affect the cumulative return difference between the rollout trajectories of DLPA and the optimal trajectories: 1. The estimation error $m$ for the discrete action $k$. 2. The estimation error $\Delta_{k,\hat{k}} + \Delta_{\omega,\hat{\omega}}$ for the distribution $\omega$ over the continuous parameters. 3. The transition and reward model estimation error $\epsilon_T$, $\epsilon_R$. It also shows how the smoothness of the transition function and reward function will affect DLPA's performance. We also provide a bound for the multi-step prediction error (compounding error) of DLPA in Appendix Theorem 8.1.

The following lemma further shows how the estimation error $\frac{m}{H}\Delta_{k,\hat{k}} + \Delta_{\omega,\hat{\omega}}$ for the continuous parameters changes with respect to the number of samples and the dimentionality of the space of continuous parameters.

**Lemma 5.3.** Let $|Z|$ denote the cardinality of the continuous parameters space and $N$ be the number of samples. Then, with probability at least $1 - \delta$:

$$\frac{m}{H}\Delta_{k,\hat{k}} + \Delta_{\omega,\hat{\omega}} \leq \frac{2m}{H}\sqrt{|Z|} + \frac{2}{N}\ln\frac{2|Z|}{\delta} \tag{9}$$

---

[1]We define the distribution over $z$ given $k$ and use it in all the following theoretical analysis to be consistent with the proposed method described in Section 4.2: for each $k$, we keep a distribution over $z$ and sample from it to roll out the trajectory.

## 6 EXPERIMENTS

We evaluated the performance of DLPA on eight standard PAMDP benchmarks, including Platform and Goal [26], Catch Point [7], Hard Goal and four versions of Hard Move. **Note that these 8 benchmarks are exactly the same environments tested in Li et al. [23]**, which introduced the algorithm (HyAR) that reaches state-of-the-art performance in these environments. We have provided a short description for each environment in appendix 8.2.

**Implementation details.** We parameterize all models using MLPs with stochastic outputs. The planning Horizon for all tasks is chosen between $\{5, 8, 10\}$. During planning, we select the trajectories with top-$\{100, 400\}$ cumulative returns, and we run 6 planning iterations. We provide more details about the hyperparameters in the appendix.

**Baselines.** We evaluate DLPA against 5 different standard PAMDP algorithms across all the 8 tasks. HyAR [23] and P-DQN [42; 3] are state-of-the-art RL algorithms for PAMDPs. HyAR learns an embedding of the parameterized action space which has been shown to be especially helpful when the dimensionality of the parameterized action space is large. P-DQN learns an actor network for every discrete action type. PADDPG [19] and HPPO [7] share similar ideas that an actor is learned to output the concatenation of discrete action and continuous parameter together. Following Li et al. [23], we replace DDPG with TD3 [9] in PADDPG and PDQN to make the comparison fair and rename PADDPG as PATD3.

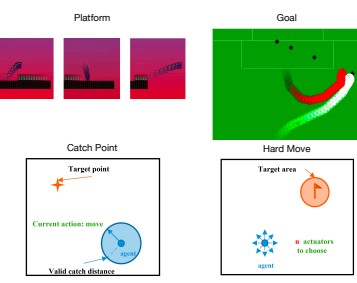

Figure 2: Visualization of the tested environments.

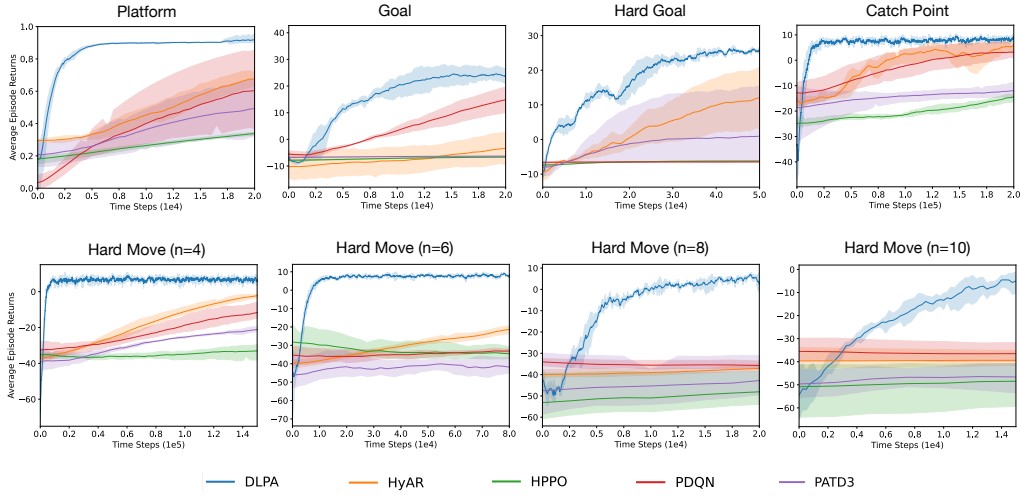

Figure 3: Comparison of different algorithms across the 8 PAMDP benchmarks. Our algorithm DLPA significantly outperforms state-of-the-art PAMDP algorithms in terms of sample efficiency. See Appendix for a full timescale version of this plot. Note that HyAR has an additional 20000 environment steps pretraining for the action encoder which we do not include in the plot.

### 6.1 RESULTS

We show the evaluation results in Figure 3 and Table 1 (for asymptotic performance). We find that DLPA achieves significantly higher sample efficiency with better or comparable asymptotic performance across all the 8 different tasks. Among all the model-free RL algorithms, HyAR achieves the overall best performance among all the other baselines, which is consistent with the results shown in their original paper. DLPA on average achieves $30\times$ **higher sample efficiency** compared to the best model-free RL method in each scenario. In all the 8 scenarios except Platform and Goal, DLPA reaches a better asymptotic performance, while in those

two domains, the final performance is still close to HyAR. In Hard Move (n= 6, 8, 10), it has been shown in HyAR's orignal paper that, no regular PAMDP algorithms can learn a meaningful policy without learning a latent action embedding space. This happens because the action space is too large (i.e., $2^n$). **However, we find that our algorithm DLPA without learning such embedding space can achieve even better performance just by sampling from the original large action space.** We can see from the table that, as the action space becomes larger (from 4 to 10), the gap between DLPA and HyAR also becomes larger. HyAR's learned action embeddings are indeed useful in these cases, but it also sacrifices computational efficiency by making the algorithm much more sophisticated. By leveraging the proposed sampling procedures during planning, DLPA as a model-based RL algorithm is able to quickly narrow down the range of candidate parameterized actions and identify the optimal ones even when the action space is extremely large.

|  | DLPA | HyAR | HPPO | PDQN | PATD3 |
|---|---|---|---|---|---|
| *Platform* | $0.92 \pm 0.05$ | $\mathbf{0.98 \pm 0.08}$ | $0.82 \pm 0.02$ | $0.91 \pm 0.07$ | $0.92 \pm 0.09$ |
| *Goal* | $28.75 \pm 6.91$ | $\mathbf{34.23 \pm 3.71}$ | $-6.17 \pm 0.06$ | $33.13 \pm 5.68$ | $-2.25 \pm 8.11$ |
| *Hard Goal* | $\mathbf{28.38 \pm 2.88}$ | $26.41 \pm 3.59$ | $-6.16 \pm 0.06$ | $1.04 \pm 10.82$ | $2.60 \pm 11.12$ |
| *Catch Point* | $\mathbf{7.56 \pm 4.86}$ | $5.20 \pm 4.18$ | $4.44 \pm 3.25$ | $6.64 \pm 2.52$ | $0.56 \pm 10.40$ |
| *Hard Move (4)* | $\mathbf{6.29 \pm 5.74}$ | $6.09 \pm 1.67$ | $-31.20 \pm 5.58$ | $4.29 \pm 4.86$ | $-10.67 \pm 3.57$ |
| *Hard Move (6)* | $\mathbf{8.48 \pm 5.45}$ | $6.33 \pm 2.12$ | $-32.21 \pm 6.54$ | $-15.62 \pm 8.65$ | $-35.50 \pm 25.43$ |
| *Hard Move (8)* | $\mathbf{7.80 \pm 6.27}$ | $-0.88 \pm 3.83$ | $-37.11 \pm 10.10$ | $-37.90 \pm 4.07$ | $-30.56 \pm 12.21$ |
| *Hard Move (10)* | $\mathbf{6.35 \pm 9.97}$ | $-7.05 \pm 3.74$ | $-39.18 \pm 8.76$ | $-39.68 \pm 5.93$ | $-43.17 \pm 15.98$ |

Table 1: Comparison of different algorithms on all the eight benchmarks at the end of training (asymptotic performance). We report the mean and standard deviation of the last ten steps before the end of training over 3 runs. Value in bold indicates the best result for each task.

## 6.2 VISUALIZATION OF PLANNING ITERATIONS.

As we mentioned before, for each planning step we run our prediction and sampling algorithm for 6 iterations and then pick the parameterized action. We show in Figure 4 the visualization of the imagined trajectories with top-30 returns for each iteration at a random time step when we evaluate DLPA in the Catch Point environment. Recall that we first sample the action sequences given a set of distribution parameters and then generate the trajectories and compute cumulative returns using the learned dynamics models. We can see that at the first iteration, the generated actions are quite random and cover a large part of the space. Then as we keep updating the distribution parameters with the cumulative returns inferred by the learned models, the imagined trajectories become more and more concentrated and finally narrow down to a relatively small-entropy distribution centering at the optimal actions. This indicates that the proposed planning method (described in Section 4.2) is able to help the agent find the greedy action to execute given the learned dynamics model while also keep a mind for exploration.

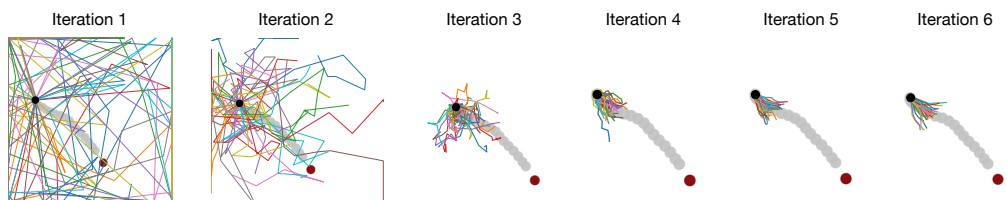

Figure 4: Visualization of DLPA's Planning iterations on the Catch Point tasks. Different color represents different imagined trajectories. The black point represents the agent's current position and the red point represents the target point. The grey trajectory is the actual trajectory taken by the agent.

## 6.3 ABLATION STUDY

In this section, we investigate the importance of some major components in DLPA: the planning algorithm, the trajectory-wise prediction for the dynamics model learning and the parameterized-action-specialized sampling distribution. We show the experimental results on *Platform* and *Goal*.

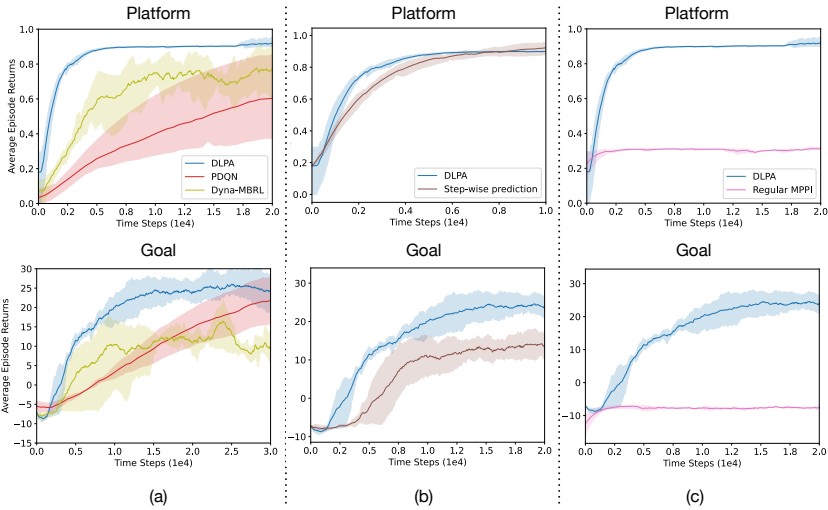

Figure 5: Ablation study on (a) the planning algorithm, (b) the trajectory-wise prediction, (c) MPPI sampling distribution.

We first investigate how the category of the planning algorithm will affect the performance of Model-based RL in PAMDP. We compare DLPA with a Dyna-like non-MPC model-based RL baseline, where we explicitly train a policy using the data generated from the learned model. As shown in Figure 5 first column, this non-MPC baseline generally performs better than the model-free baselines but is not as good as DLPA.

As we mentioned in Section 4, an important difference between our method and many prior model-based RL algorithms is that, when updating the dynamics model, we only give the model the start state and the action sequences sampled from the replay buffer as input and let it predict the whole trajectory. In contrast, a common way to do the inference during the training of dynamics model is just give the model every ground-truth states and actions, then let it predict the next state and reward and compute the loss. We show an empirical performance comparison for these two ways of updating the dynamics models in Figure 5 second column. DLPA achieves significantly higher sample efficiency by predicting into several steps into the future with a length of horizon $H$. Presumably this is because during planning we will plan into the future with the exact same length $H$ thus the proposed updating process will help the agent to focus on predicting the parts of state more accurately which will affect the future more and help the agent achieve better cumulative return.

Finally, the other major modification we make in the CEM planning process is that we keep a separate distribution over the continuous parameters for each discrete action and update them at each iteration instead of keeping just one independent distribution for the discrete actions and one independent distribution for the continuous parameters. We investigate the influence of this change by comparing to just a version of DLPA that just uses one independent distiribution for all the continuous parameters. The results are shown in Figure 5 third column,without this technique it is quite hard for DLPA to do proper planning.

## 7  CONCLUSION

We have introduced DLPA, the first model-based Reinforcement Learning algorithm for parameterized action spaces (also known as discrete-continuous hybrid action spaces). DLPA first learns a dynamics model that is conditioned on the parameterized actions with a weighted trajectory-level prediction loss. Then we propose a novel planning method for parameterized actions by keep updating and sampling from the distribution over the discrete actions and continuous parameters. DLPA outperforms the state-of-the-art PAMDP algorithms on 8 standard PAMDP benckmarks. We further empirically demonstrate the effectiveness of the different components of the proposed algorithm.

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

## 8 APPENDIX

### 8.1 MORE THEORETICAL RESULTS AND PROOFS

**Theorem 8.1.** *For a $(L_R^S, L_R^K, L_R^Z, L_T^S, L_T^K, L_T^Z)$-Lipschitz PAMDP and the learned DLPA $\epsilon_T$-accurate transition model $\hat{T}$ and $\epsilon_R$-accurate reward model $\hat{T}$, let $q(s)$ be the initial state distribution, $L_{\hat{T}}^S = \min\{L_T^S, L_{\hat{T}}^S\}$, $L_{\hat{T}}^K = \min\{L_T^K, L_{\hat{T}}^K\}$, $L_{\hat{T}}^Z = \min\{L_T^Z, L_{\hat{T}}^Z\}$ and similarly define $L_{\hat{R}}^S$, $L_{\hat{R}}^K$, $L_{\hat{R}}^Z$. Then $\forall n > 1$, starting from the same initial distribution $q(s)$, the n-step prediction error is bounded by:*

$$
\begin{aligned}
\Delta(n) :=&W\Big(T_n(q(s), k, \omega(\cdot|k)), \hat{T}_n(q(s), \hat{k}, \hat{\omega}(\cdot|\hat{k}))\Big) \\
\leq &\ (\epsilon_T)^n + (L_{\hat{T}}^Z \Delta_{\omega,\hat{\omega}} + L_{\hat{T}}^Z \Delta_{k,\hat{k}}) \sum_{i=0}^{n-1}(L_{\hat{T}}^S)^i + L_{\hat{T}}^K \sum_{i=0}^{n-1}(L_{\hat{T}}^S)^i \mathbb{1}(k_{n-i} \neq \hat{k_{n-i}})
\end{aligned}
\tag{10}
$$

*Proof.* Firstly, for one-step prediction given the initial distribution $q(s)$ and the same $k$ as well as $\omega$:

$$
\begin{aligned}
\Delta(1) :=&W\Big(T(q(s), k, \omega(\cdot|k)), \hat{T}(q(s), k, \omega(\cdot|k))\Big) \\
=&\sup_f \int \int (\hat{T}(s'|s, k, \omega) - T(s'|s, k, \omega))f(s')q(s)dsds' \textbf{ Duality for Wasserstein Metric} \\
\leq&\int \sup_f \int (\hat{T}(s'|s, k, \omega) - T(s'|s, k, \omega))f(s')ds'q(s)ds \textbf{ Jensen's inequality} \\
=&\int W(\hat{T}(s'|s, k, \omega), T(s'|s, k, \omega))q(s)ds \leq \int \epsilon_T q(s)ds = \epsilon_T
\end{aligned}
\tag{11}
$$

Then for the n-step prediction error with different discrete action and continuous parameter distributions:

$$
\begin{aligned}
\Delta(n) :=&W\Big(T_n(q(s), k, \omega(\cdot|k)), \hat{T}_n(q(s), \hat{k}, \hat{\omega}(\cdot|\hat{k}))\Big) \\
\leq&\ W\Big(T_n(q(s), k, \omega(\cdot|k)), T_n(q(s), \hat{k}, \hat{\omega}(\cdot|\hat{k}))\Big) + W\Big(T_n(q(s), \hat{k}, \hat{\omega}(\cdot|\hat{k})), \hat{T}_n(q(s), \hat{k}, \hat{\omega}(\cdot|\hat{k}))\Big) \textbf{ Triangle inequality}
\end{aligned}
\tag{12}
$$

For the second term in 12:

$$
\begin{aligned}
W\Big(T_n(q(s), \hat{k}, \hat{\omega}(\cdot|\hat{k})), \hat{T}_n(q(s), \hat{k}, \hat{\omega}(\cdot|\hat{k}))\Big) = &W\Big(T(T_{n-1}(q(s), \hat{k}, \hat{\omega}(\cdot|\hat{k})), \hat{k}_n, \hat{\omega}_n), \hat{T}(\hat{T}_{n-1}(q(s), \hat{k}, \hat{\omega}(\cdot|\hat{k})), \hat{k}_n, \hat{\omega}_n)\Big) \\
\leq &\epsilon_T W\Big(T_{n-1}(q(s), \hat{k}, \hat{\omega}(\cdot|\hat{k})), \hat{T}_{n-1}(q(s), \hat{k}, \hat{\omega}(\cdot|\hat{k}))\Big) \textbf{ Composition Lemma [2]} \\
&\cdots \\
\leq &(\epsilon_T)^n
\end{aligned}
\tag{13}
$$

For the first term in 12

$$
\begin{aligned}
&W\Big(T_n(q(s), k, \omega(\cdot|k)), T_n(q(s), \hat{k}, \hat{\omega}(\cdot|\hat{k}))\Big) \leq \\
&W\Big(T_n(q(s), k, \omega(\cdot|k)), T_n(q(s), k, \hat{\omega}(\cdot|\hat{k}))\Big) + W\Big(T_n(q(s), k, \hat{\omega}(\cdot|\hat{k})), T_n(q(s), \hat{k}, \hat{\omega}(\cdot|\hat{k}))\Big)
\end{aligned}
\tag{14}
$$

For the second term in 14:

$$W\Big(T_n(q(s), k, \hat{\omega}(\cdot|\hat{k})), T_n(q(s), \hat{k}, \hat{\omega}(\cdot|\hat{k}))\Big)$$

$$= W\Big(T(T_{n-1}(q(s), k, \hat{\omega}(\cdot|\hat{k})), k_n, \hat{\omega_n}), T(T_{n-1}(q(s), \hat{k}, \hat{\omega}(\cdot|\hat{k})), \hat{k_n}, \hat{\omega_n})\Big)$$

$$\le L_T^K d(k_n, \hat{k_n}) + L_T^S W\Big(T_{n-1}(q(s), k, \hat{\omega}(\cdot|\hat{k})), T_{n-1}(q(s), \hat{k}, \hat{\omega}(\cdot|\hat{k}))\Big)$$

$$= L_T^K \mathbb{1}(k_n \ne \hat{k_n}) + L_T^S W\Big(T_{n-1}(q(s), k, \hat{\omega}(\cdot|\hat{k})), T_{n-1}(q(s), \hat{k}, \hat{\omega}(\cdot|\hat{k}))\Big)$$

$$\le L_T^K \mathbb{1}(k_n \ne \hat{k_n}) + L_T^S L_T^K \mathbb{1}(k_{n-1} \ne \hat{k_{n-1}}) + (L_T^S)^2 W\Big(T_{n-2}(q(s), k, \hat{\omega}(\cdot|\hat{k})), T_{n-2}(q(s), \hat{k}, \hat{\omega}(\cdot|\hat{k}))\Big)$$

$$\cdots$$

$$\le L_T^K \sum_{i=0}^{n-1} (L_T^S)^i \mathbb{1}(k_{n-i} \ne \hat{k_{n-i}}) \tag{15}$$

For the first term in 14:

$$W\Big(T_n(q(s), k, \omega(\cdot|k)), T_n(q(s), k, \hat{\omega}(\cdot|\hat{k}))\Big) \le$$

$$W\Big(T_n(q(s), k, \omega(\cdot|k)), T_n(q(s), k, \hat{\omega}(\cdot|k))\Big) + W\Big(T_n(q(s), k, \hat{\omega}(\cdot|k)), T_n(q(s), k, \hat{\omega}(\cdot|\hat{k}))\Big) \tag{16}$$

For the first term in 16:

$$W\Big(T_n(q(s), k, \omega(\cdot|k)), T_n(q(s), k, \hat{\omega}(\cdot|k))\Big)$$

$$= W\Big(T(T_{n-1}(q(s), k, \omega(\cdot|k)), k_n, \omega_n), T(T_{n-1}(q(s), k, \hat{\omega}(\cdot|k)), k_n, \hat{\omega_n})\Big)$$

$$\le L_T^Z \Delta_{\omega, \hat{\omega}} + L_T^S W\Big(T_{n-1}(q(s), k, \omega(\cdot|k)), T_{n-1}(q(s), k, \hat{\omega}(\cdot|k))\Big)$$

$$\le L_T^Z \Delta_{\omega, \hat{\omega}} + L_T^S L_T^Z \Delta_{\omega, \hat{\omega}} + (L_T^S)^2 W\Big(T_{n-2}(q(s), k, \omega(\cdot|k)), T_{n-2}(q(s), k, \hat{\omega}(\cdot|k))\Big)$$

$$\cdots$$

$$\le L_T^Z \Delta_{\omega, \hat{\omega}} \sum_{i=0}^{n-1} (L_T^S)^i \tag{17}$$

Similarly, for the second term in 16:

$$W\Big(T_n(q(s), k, \hat{\omega}(\cdot|k)), T_n(q(s), k, \hat{\omega}(\cdot|\hat{k}))\Big)$$

$$\le L_T^Z \Delta_{k, \hat{k}} \sum_{i=0}^{n-1} (L_T^S)^i \mathbb{1}(k_{n-i} \ne \hat{k_{n-i}}) \tag{18}$$

Combining all the results above and continue 12, we have:

$$\Delta(n) \le (\epsilon_T)^n + L_T^K \sum_{i=0}^{n-1} (L_T^S)^i + L_T^Z \Delta_{\omega, \hat{\omega}} \sum_{i=0}^{n-1} (L_T^S)^i + L_T^Z \Delta_{k, \hat{k}} \sum_{i=0}^{n-1} (L_T^S)^i$$

$$= (\epsilon_T)^n + \sum_{i=0}^{n-1} (L_T^S)^i (L_T^Z \Delta_{\omega, \hat{\omega}} + L_T^Z \Delta_{k, \hat{k}} \mathbb{1}(k_{n-i} \ne \hat{k_{n-i}})) + L_T^K \sum_{i=0}^{n-1} (L_T^S)^i \mathbb{1}(k_{n-i} \ne \hat{k_{n-i}}) \tag{19}$$

Now if replace $T_n(q(s), \hat{k}, \hat{\omega}(\cdot|\hat{k}))$ with $\hat{T}_n(q(s), k, \omega(\cdot|k))$ in the triangle inequality 12 and do all the derivation again, we have:

$$\Delta(n) \le (\epsilon_T)^n + \sum_{i=0}^{n-1} (L_{\hat{T}}^S)^i (L_{\hat{T}}^Z \Delta_{\omega, \hat{\omega}} + (L_{\hat{T}}^Z \Delta_{k, \hat{k}} + L_{\hat{T}}^K) \mathbb{1}(k_{n-i} \ne \hat{k_{n-i}})) \tag{20}$$

Combining 19 and 20 concludes the proof. $\qquad\square$

**Theorem 5.2.** For a $(L_R^S, L_R^K, L_R^Z, L_T^S, L_T^K, L_T^Z)$-Lipschitz PAMDP and the learned DLPA $\epsilon_T$-accurate transition model $\hat{T}$ and $\epsilon_R$-accurate reward model $\hat{T}$, let $L_{\bar{T}}^S = \min\{L_T^S, L_{\hat{T}}^S\}$, $L_{\bar{T}}^K = \min\{L_T^K, L_{\hat{T}}^K\}$, $L_{\bar{T}}^Z = \min\{L_T^Z, L_{\hat{T}}^Z\}$ and similarly define $L_{\bar{R}}^S, L_{\bar{R}}^K, L_{\bar{R}}^Z$. If $L_{\bar{T}}^S < 1$, then the regret of the rollout trajectory $\hat{\tau} = \{\hat{s_1}, \hat{k_1}, \hat{\omega}_1(\cdot|\hat{k_1}), \hat{s_2}, \hat{k_2}, \hat{\omega}_2(\cdot|\hat{k_2}), \cdots\}$ from DLPA is bounded by:

$$|\mathcal{J}_{\tau^*} - \mathcal{J}_{\hat{\tau}}| := \sum_{t=1}^{H} \gamma^{t-1} |R(s_t, k_t, \omega_t(\cdot|k_t)) - \hat{R}(\hat{s_t}, \hat{k_t}, \hat{\omega}_t(\cdot|\hat{k_t}))|$$
$$\leq \mathcal{O}\Big((L_{\bar{R}}^K + L_{\bar{R}}^S L_{\bar{T}}^K)m + H(\epsilon_R + L_{\bar{R}}^S \epsilon_T + (L_{\bar{R}}^Z + L_{\bar{R}}^S L_{\bar{T}}^Z)(\frac{m}{H}\Delta_{k,\hat{k}} + \Delta_{\omega,\hat{\omega}}))\Big),$$
$$\tag{21}$$

where $m = \sum_{t=1}^{H} \mathbb{1}(k_t \neq \hat{k}_t), \Delta_{\omega,\hat{\omega}} = W(\omega(\cdot|k), \hat{\omega}(\cdot|k)), \Delta_{k,\hat{k}} = W(\omega(\cdot|k), \omega(\cdot|\hat{k})), N$ denotes the number of samples and $H$ is the planning horizon.

*Proof.* At timestep $t$:

$$|R(s_t, k_t, \omega_t(\cdot|k_t)) - \hat{R}(\hat{s_t}, \hat{k_t}, \hat{\omega}_t(\cdot|\hat{k_t}))| \leq$$
$$|R(s_t, k_t, \omega_t(\cdot|k_t)) - \hat{R}(\hat{s_t}, k_t, \hat{\omega}_t(\cdot|\hat{k_t}))| + |\hat{R}(\hat{s_t}, k_t, \hat{\omega}_t(\cdot|\hat{k_t})) - \hat{R}(\hat{s_t}, \hat{k_t}, \hat{\omega}_t(\cdot|\hat{k_t}))| \tag{22}$$

For the second term in 22:

$$|\hat{R}(\hat{s_t}, k_t, \hat{\omega}_t(\cdot|\hat{k_t})) - \hat{R}(\hat{s_t}, \hat{k_t}, \hat{\omega}_t(\cdot|\hat{k_t}))| \leq L_{\hat{R}}^K d(k, \hat{k}) = L_{\hat{R}}^K \mathbb{1}(k_t \neq \hat{k}_t) \tag{23}$$

For the first term in 22:

$$|R(s_t, k_t, \omega_t(\cdot|k_t)) - \hat{R}(\hat{s_t}, k_t, \hat{\omega}_t(\cdot|\hat{k_t}))| \leq$$
$$|R(s_t, k_t, \omega_t(\cdot|k_t)) - \hat{R}(\hat{s_t}, k_t, \omega_t(\cdot|k_t))| + |\hat{R}(\hat{s_t}, k_t, \omega_t(\cdot|k_t)) - \hat{R}(\hat{s_t}, k_t, \hat{\omega}_t(\cdot|\hat{k_t}))|$$
$$\leq \epsilon_R + L_{\hat{R}}^S \Delta(t-1) + |\hat{R}(\hat{s_t}, k_t, \omega_t(\cdot|k_t)) - \hat{R}(\hat{s_t}, k_t, \hat{\omega}_t(\cdot|\hat{k_t}))| \text{ By the definition of } \Delta(n) \text{ and Composition Lemma}$$
$$\leq \epsilon_R + L_{\hat{R}}^S \Delta(t-1) + |\hat{R}(\hat{s_t}, k_t, \omega_t(\cdot|k_t)) - \hat{R}(\hat{s_t}, k_t, \omega_t(\cdot|\hat{k_t}))| + |\hat{R}(\hat{s_t}, k_t, \omega_t(\cdot|\hat{k_t})) - \hat{R}(\hat{s_t}, k_t, \hat{\omega}_t(\cdot|\hat{k_t}))|$$
$$\leq \epsilon_R + L_{\hat{R}}^S \Delta(t-1) + L_{\hat{R}}^Z \Delta_{k,k'} + |\hat{R}(\hat{s_t}, k_t, \omega_t(\cdot|\hat{k_t})) - \hat{R}(\hat{s_t}, k_t, \hat{\omega}_t(\cdot|\hat{k_t}))|$$
$$\leq \epsilon_R + L_{\hat{R}}^S \Delta(t-1) + L_{\hat{R}}^Z (\Delta_{k,\hat{k}} + \Delta_{\omega,\hat{\omega}})$$
$$\leq \epsilon_R + L_{\hat{R}}^S (\epsilon_T)^{t-1} + L_{\hat{R}}^S \sum_{i=0}^{t-2} (L_{\hat{T}}^S)^i (L_{\hat{T}}^Z \Delta_{\omega,\hat{\omega}} + (L_{\hat{T}}^Z \Delta_{k,\hat{k}} + L_{\hat{T}}^K) \mathbb{1}(k_{t-1-i} \neq \hat{k_{t-1-i}}))$$
$$+ L_{\hat{R}}^Z (\Delta_{k,\hat{k}} + \Delta_{\omega,\hat{\omega}}) \text{ According to Theorem 8.1}$$
$$\tag{24}$$

Now we go back to 22:

$$|R(s_t, k_t, \omega_t(\cdot|k_t)) - \hat{R}(\hat{s_t}, \hat{k_t}, \hat{\omega}_t(\cdot|\hat{k_t}))| \leq$$
$$|R(s_t, k_t, \omega_t(\cdot|k_t)) - \hat{R}(\hat{s_t}, k_t, \hat{\omega}_t(\cdot|\hat{k_t}))| + |\hat{R}(\hat{s_t}, k_t, \hat{\omega}_t(\cdot|\hat{k_t})) - \hat{R}(\hat{s_t}, \hat{k_t}, \hat{\omega}_t(\cdot|\hat{k_t}))|$$
$$\leq \epsilon_R + L_{\hat{R}}^K \mathbb{1}(k_t \neq \hat{k}_t) + L_{\hat{R}}^S (\epsilon_T)^{t-1} + L_{\hat{R}}^S \sum_{i=0}^{t-2} (L_{\hat{T}}^S)^i (L_{\hat{T}}^Z \Delta_{\omega,\hat{\omega}} + (L_{\hat{T}}^Z \Delta_{k,\hat{k}} + L_{\hat{T}}^K) \mathbb{1}(k_{t-1-i} \neq \hat{k_{t-1-i}}))$$
$$+ L_{\hat{R}}^Z (\Delta_{k,\hat{k}} + \Delta_{\omega,\hat{\omega}})$$
$$\tag{25}$$

Then we can compute the regret:

$$
\begin{aligned}
|\mathcal{J}_{\tau^*} - \mathcal{J}_{\hat{\tau}}| &:= \sum_{t=1}^{H} \gamma^{t-1} |R(s_t, k_t, \omega_t(\cdot|k_t)) - \hat{R}(\hat{s}_t, \hat{k}_t, \hat{\omega}_t(\cdot|\hat{k}_t))| \\
&\leq \sum_{t=1}^{H} \gamma^{t-1} [\epsilon_R + L_{\hat{R}}^K \mathbb{1}(k_t \neq \hat{k}_t) + L_{\hat{R}}^S (\epsilon_T)^{t-1} + \\
&\quad L_{\hat{R}}^S \sum_{i=0}^{t-2} (L_{\hat{T}}^S)^i (L_{\hat{T}}^Z \Delta_{\omega,\hat{\omega}} + (L_{\hat{T}}^Z \Delta_{k,\hat{k}} + L_{\hat{T}}^K) \mathbb{1}(k_{t-1-i} \neq \hat{k_{t-1-i}})) + L_{\hat{R}}^Z (\Delta_{k,\hat{k}} + \Delta_{\omega,\hat{\omega}})]
\end{aligned}
$$

$$
\leq \mathcal{O}\Big( (L_{\hat{R}}^K + L_{\hat{R}}^S L_{\hat{T}}^K) m + H(\epsilon_R + L_{\hat{R}}^S \epsilon_T + (L_{\hat{R}}^Z + L_{\hat{R}}^S L_{\hat{T}}^Z)(\frac{m}{H} \Delta_{k,\hat{k}} + \Delta_{\omega,\hat{\omega}})) \Big) \text{ \textbf{Assuming} } \gamma = 1
\tag{26}
$$

Similar to the proof of theorem 8.1, if we change the middle term in the triangle inequalities we will have:

$$
|\mathcal{J}_{\tau^*} - \mathcal{J}_{\hat{\tau}}| \leq \mathcal{O}\Big( (L_{\hat{R}}^K + L_{\hat{R}}^S L_{\hat{T}}^K) m + H(\epsilon_R + L_{\hat{R}}^S \epsilon_T + (L_{\hat{R}}^Z + L_{\hat{R}}^S L_{\hat{T}}^Z)(\frac{m}{H} \Delta_{k,\hat{k}} + \Delta_{\omega,\hat{\omega}})) \Big)
\tag{27}
$$

Combine 26 and 27 concludes the proof. □

**Lemma 8.2.** *Let $|Z|$ denote the cardinality of the continuous parameters space and $N$ be the number of samples. Then, with probability at least $1 - \delta$:*

$$
\frac{m}{H} \Delta_{k,\hat{k}} + \Delta_{\omega,\hat{\omega}} \leq \frac{2m}{H} \sqrt{|Z|} + \frac{2}{N} \ln \frac{2|Z|}{\delta}
\tag{28}
$$

*Proof.* By definition:

$$
\Delta_{k,\hat{k}} := W(\omega(\cdot|k), \omega(\cdot|\hat{k}))
\tag{29}
$$

Recall that in our algorithm, we assume $\omega(\cdot|k)$ is a Gaussian distribution $\mathcal{N}(\mu_k, \Sigma_k)$ and $z \sim \omega(\cdot)$ takes value in the range $[-1, 1]$.

Now we use the definition of **2nd Wasserstein distance** $W_2$ between multivariate Gaussian distributions:

$$
W_2(\omega(\cdot|k), \omega(\cdot|\hat{k})) = \|\mu_k - \mu_{\hat{k}}\|_2^2 + Tr(\Sigma + \hat{\Sigma} - 2(\Sigma^{1/2} \hat{\Sigma} \Sigma^{1/2})^{1/2})
\tag{30}
$$

Ignore the covariance term, we have:

$$
W(\omega(\cdot|k), \omega(\cdot|\hat{k})) \leq 2\sqrt{|Z|}
\tag{31}
$$

By definition:

$$
\Delta_{\omega,\hat{\omega}} := W(\omega(\cdot|k), \hat{\omega}(\cdot|k))
\tag{32}
$$

Similarly, we have:

$$
W_2(\omega(\cdot|k), \hat{\omega}(\cdot|k)) = \|\mu_k - \hat{\mu_k}\|_2^2 + Tr(\Sigma + \hat{\Sigma} - 2(\Sigma^{1/2} \hat{\Sigma} \Sigma^{1/2})^{1/2})
\tag{33}
$$

Ignore the covariance term, by **Hoeffding's inequality**, with probability $1 - \delta$ we have for each dimension $i$ of $Z$:

$$
\|\mu_{k,i} - \hat{\mu_{k,i}}\|_2^2 \leq \frac{(z_i^{max} - z_i^{min})^2}{2N} \ln \frac{2}{\delta}
\tag{34}
$$

By the union bound and the range of $z$:

$$
\|\mu_k - \hat{\mu_k}\|_2^2 \leq \frac{2}{N} \ln \frac{2|Z|}{\delta}
\tag{35}
$$

Combining the results, we have:

$$
\frac{m}{H} Delta_{k,\hat{k}} + \Delta_{\omega,\hat{\omega}} = \frac{m}{H} W_2(\omega(\cdot|k), \omega(\cdot|\hat{k})) + W_2(\omega(\cdot|k), \hat{\omega}(\cdot|k)) \leq \frac{2m}{H} \sqrt{|Z|} + \frac{2}{N} \ln \frac{2|Z|}{\delta}
\tag{36}
$$

□

## 8.2 ENVIRONMENT DESCRIPTION

- Platform: The agent is expected to reach the final goal while avoiding an enemy, or leaping over a gap. There are three parameterized actions (run, hop and leap) and each discrete action has one continuous parameter.

- Goal: The agent needs to find a way to avoid the goal keeper and shoot the ball into the gate. There are three parameterized actions: kick-to(x,y), shoot-goal-left(h), shoot-goal-right(h).

- Hard Goal: A more challenging version of the Goal environment where there are ten parameterized actions.

- Catch Point: The agent is expected to catch a goal point within limited trials. There are two parameterized actions: Move(d), catch(d).

- Hard Move (n= 4, 6, 8, 10): This is a set of environments where the agent needs to control $n$ actuators to reach a goal point. The number of parameterized actions is $2^n$.

## 8.3 NETWORK STRUCTURE

**We use the official code provided by HyAR[2] to implement all the baselines on the 8 benchmarks.** The dynamics models in our proposed DLPA consists of three components: transition predictor $T_\phi$, continue predictor $p_\phi$ and reward predictor $R_\phi$, whose structures are shown in Table 8.3 and the hyperparameters are shown in Table 3.

| Layer | Transition Predictor | Continue Predictor | Reward Predictor |
|---|---|---|---|
| Fully Connected | $(inp\_dim, 64)$ | $(inp\_dim, 64)$ | $(inp\_dim, 64)$ |
| Activation | ReLU | ReLU | ReLU |
| Fully Connected | $(64, 64)$ | $(64, 64)$ | $(64, 64)$ |
| Activation | ReLU | ReLU | ReLU |
| Fully Connected | $(64, state\_dim)$ | $(64, 2)$ | $(64, 1)$ |
| Fully Connected | $(64, state\_dim)$ | $(64, 2)$ | $(64, 1)$ |

Table 2: Network structures for all three predictors, $inp\_dim$ is the size of state space, discrete action space and continuous parameter space. Instead of outputting a deterministic value, our networks output parameters of a Gaussian distribution, which are mean and log standard deviation.

| Hyperparameter | Value |
|---|---|
| Discount factor($\gamma$) | 0.99 |
| Horizon | 10, 8, 8, 5, 5, 5, 5, 5 |
| Replay buffer size | $10^6$ |
| Population size | 1000 |
| Elite size | 400 |
| CEM iteration | 6 |
| Temperature | 0.5 |
| Learning rate | 3e-4 |
| Transition loss coefficient | 1 |
| Reward loss coefficient | 0.5 |
| Termination loss coefficient | 1 |
| Batch size | 128 |
| Steps per update | 1 |

Table 3: DLPA hyperparameters. We list the most important hyperparameters during both training and evaluating. If there's only one value in the list, it means all environments use the same value, otherwise, it's in the order of Platform, Goal, Hard Goal, Catch Point, Hard Move (4), Hard Move (6), Hard Move (8), and Hard Move (10).

It is worth noting that we train two reward predictors each time in Hard Move and Catch Point environments. Conditioned on whether the prediction for termination is True or False, we train one

---

[2]https://github.com/TJU-DRL-LAB/self-supervised-rl/tree/main/RL_with_
Action_Representation/HyAR

reward prediction network to only predict the reward when the trajectory has not terminated, and one network for the case when the prediction from the continue predictor is True.

## 8.4 COMPLETE LEARNING CURVES

We provide the full timescale plot of the training performance comparison on the 8 PAMDP benchmarks in Fig. 6. In general, the proposed method DLPA achieves significantly better sample efficiency and asymptotic performance than all the state-of-the-art PAMDP algorithms in most scenarios.

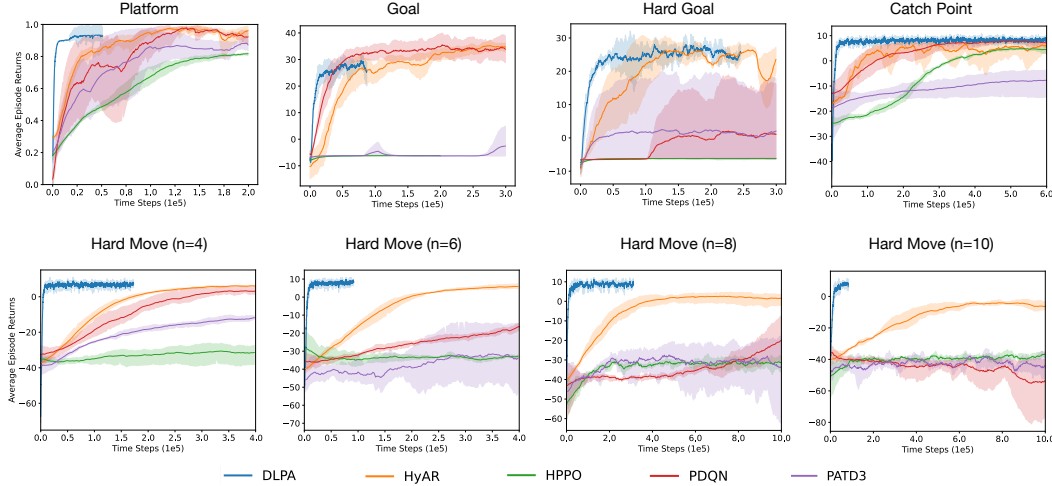

Figure 6: Comparison of different algorithms across the 8 PAMDP benchmarks, when the model-free methods converges. Our algorithm DLPA stops early in the experiments because it already converges.

## 8.5 ADDITIONAL ABLATION STUDY

Next we conduct an ablation study on the planning algorithm. In Section 4.2, we design a special sampling and updating algorithm for parameterized action spaces, here we compare it with a method that just randomly samples from a fixed distribution and picks the best action at every time step (also known as "random shooting"). The results are shown in Figure 8.5. The proposed method DLPA significantly outperforms the version of the algorithm that uses random shooting to do sampling and get the optimal actions. Parameterized action space in general is a much larger sampling space, comparing to just discrete or continuous action space. This is because each time the agent need to first sample the discrete actions and each discrete action has a independent continuous parameter space. The problem becomes more severe when the number of discrete actions is extremely large. Thus it is hard for a random-shooting method to consistently find the optimal action distributions while also augment exploration.

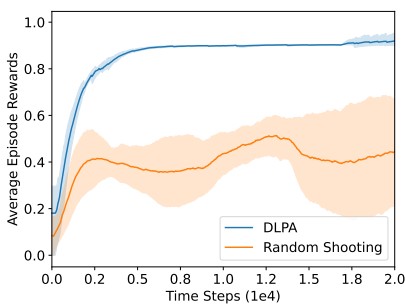

Figure 7: Ablation study on the planning algorithm (Platform).

## 8.6 COMPUTATIONAL COMPLEXITY

We also compare the clock time of planning and number of timesteps needed to converge as the action space expands. We tested this on the Hard Move domain, where the number of discrete actions changes from $2^4$ to $2^{10}$ (one continuous parameter for each of them). As shown in table 8.6, while the number of samples increases as the action space expands, it's still within an acceptable range even when it's extremely large. The results are also consistent with our theoretical analysis.

| # Discrete actions | Planning clock time /s | Training clock time /s | # Timesteps to converge |
|:---:|:---:|:---:|:---:|
| $2^4$ | 1.71e-1 | 6.99e-3 | 6,000 |
| $2^6$ | 3.05e-1 | 7.18e-3 | 8,500 |
| $2^8$ | 8.12e-1 | 7.53e-3 | 15,000 |
| $2^{10}$ | 2.85 | 7.81e-3 | 23,000 |

Table 4: Computational complexity study. We evaluate the number of timesteps needed to converge as the action space expands on the Hard Move domain.

## 8.7 ADDITIONAL EXPERIMENTS ON HFO

We further test our method on a much more complex domain—Half-Field-Offense (HFO) [18], where both the state space and action space are much larger than the 8 benchmarks. Besides, the task horizon is $10\times$ longer and there is more randomness existing in the environment. HFO is originally a subtask in RoboCup simulated soccer[3]. As shown in Figure 8, DLPA is still able to reach a better performance than all the model-free PAMDP RL baselines in this higher dimensional domain.

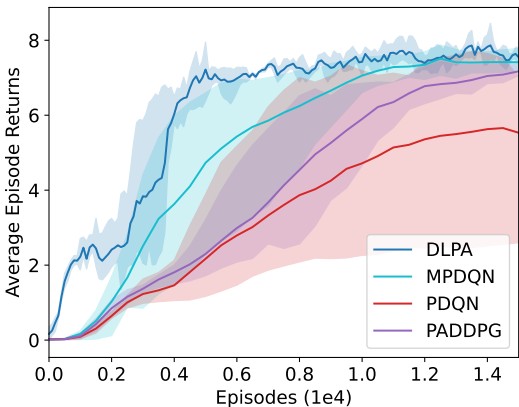

Figure 8: Additional experimental results on HFO

---

[3] https://www.robocup.org/leagues/24

