# OpenReview forum: "Model-based Reinforcement Learning for Parameterized Action Spaces"
_ICLR.cc/2024/Conference — Submitted to ICLR 2024_

### Official Review · Reviewer_75cb · 2023-10-17

**Soundness:** 3 good
**Presentation:** 2 fair
**Contribution:** 3 good
**Rating:** 5
**Confidence:** 3

**Summary:**

In this paper, the authors present the first model-based solution, DLPA, for the RL problem in PAMDP settings where both discrete and continuous actions coexist. DLPA initially learns an environment model that accommodates both discrete and continuous actions. It optimizes this model using prediction errors over a continuous H-step horizon when the state input includes only the true state at the first step. Subsequently, DLPA employs the CEM method extended to the PAMDP setting, based on this model, to perform MPC.

**Strengths:**

1. This paper introduces a model-based approach for the first time in the context of PAMDP settings and addresses the learning challenges posed by the large action space through the application of CEM-based MPC methods.
2. The experimental results showcase the superior performance of the approach presented in this paper in terms of sample efficiency and final performance.

**Weaknesses:**

1. Quite a few grammar errors：
    - Sec 1, first paragraph: “… at each time step the agent must choose a discrete action type (move, ribble or shoot) and also continuous parameters (related to) that chosen discrete action, … each discrete action is parameterized by some continuous parameters, …”.
    - Sec 1, second paragraph: “By contrast, in continuous/discrete-only action spaces, …”.
    - Sec 1, third paragraph: ”It then performs Model Predictive Control …”.
    - Sec 4.2, third paragraph: “The discrete actions and continuous parameters …”.
    - Sec 4.2, fourth paragraph: “This is because each time, the agent needs to first sample from the discrete actions then sample from the corresponding continuous parameters. And each discrete action has a independent continuous parameter space. This becomes …”.
    - Sec 4.2, fifth paragraph: “… the agent executes E steps of forward planning while updating the distribution parameters over discrete actions and continuous parameters. Then it uses the first action …”.
    - Sec 5, first paragraph: “Goal: The agent needs …”.
    - Sec 5, first paragraph: “The number of parameterized actions is …”.
    - Sec 5, second paragraph: “HyAR learns an embedding of the parameterized action space …”.
    - Sec 5.1, first paragraph: “We find that DLPA achieves significantly higher sample efficiency …, … which is consistent with the results …, … DLPA on average achieves 30× higher sample efficiency compared to the best model-free method …”.
2. The author's description of the key innovations in this paper, such as the calculation of the H-step loss in model learning or the introduction of the CEM method, does not place enough emphasis on what makes these approaches unique when applied to the specific problem scenario PAMDP. For instance, what sets apart the H-step loss in DLPA compared to the traditional model-based approaches, or how does this paper's use of CEM bring innovation in the context of PAMDP, especially regarding handling the relationship between discrete and continuous actions? These are the questions that are more central to the paper and that I would like to have a better understanding of.
3. During the experimental process, the number of training steps for DLPA differs from the other baseline methods. This discrepancy makes it challenging to adequately assess the stability of DLPA's training, including whether it exhibits oscillations or other issues. Additionally, it appears that HyAR may not achieve full convergence in certain environments. Therefore, it would be advisable to provide additional experimental results with a consistent number of training steps for all algorithms to ensure a fair comparison.
4. For the experiments assessing the impact of CEM-based MPC on DLPA, it would be beneficial to include ablation studies with other traditional model-based non-MPC methods, such as MBPO, applied in the context of PAMDP.

**Questions:**

1. What specific form do the parameters of CEM take in this context? Are they similar to the parameters of a policy neural network, and how do they relate to the state input?
2. How does the computational complexity of CEM change as the action space expands?

---

> ### Author Response · Authors · 2023-11-20
> **Official Comment by Authors (Part 1)**
>
> Thank you for the insightful feedback on our submission. We believe we can address your concerns. We have fixed the grammar errors accordingly in our updated manuscript.
>
> Q1: “The author's description of the key innovations in this paper, such as the calculation of the H-step loss in model learning or the introduction of the CEM method, does not place enough emphasis on what makes these approaches unique when applied to the specific problem scenario PAMDP.”
>
> A1: We believe the primary contribution of this paper is the very first model-based RL algorithm for the PAMDP setting. It outperforms all the previous model-free RL methods - many of which have been published at ICLR, including one just last year - for PAMDP on 8 standard PAMDP benchmarks. The PAMDP setting is very common in real-world tasks, and has drawn a lot of attention in recent papers, but all of them are model-free methods.  We hope our paper can draw other researchers' attention to the model-based aspect of this problem setting since DLPA performs better than even the most sophisticated model-free methods. Calculation of the H-step loss in model learning is an important technique that we found to be especially useful in practice. Additionally, for the CEM method in PAMDPs, previous works have only shown its efficiency on either discrete action space or continuous action space, while we’re the first to apply such a method to the parameterized action space. Note that **one major modification we make when applying MPPI to PAMDPs is that we keep a separate distribution over the continuous parameters for each discrete action and update them at each iteration instead of keeping just one independent distribution for the discrete actions and one independent distribution for the continuous parameters.** In other words, we let the distribution of the continuous parameters condition on the chosen discrete action during the sampling process. We believe this is an important change to make when using MPPI for PAMDPs as we don’t want to throw away the established dependency between the continuous parameter and corresponding discrete action in PAMDPs. In the updated manuscript Section (6.3) we have included a new ablation study result for this change which shows that this change is indispensable for DLPA to do proper planning in PAMDP.
>
> In terms of novelty, **we have also added an entirely new theoretical analysis section in the updated manuscript where we derive some performance guarantees / bounds for the proposed algorithm (Section 5).** In general, we provide three main theorems that describe the performance guarantees of DLPA from three aspects: 1. A theoretical bound for the cumulative return difference between the rollout trajectories of DLPA and the optimal trajectories. 2. A theoretical bound for the multi-step prediction error of DLPA. 3. A theoretical bound for how the estimation error will change with respect to the number of samples and the cardinality of the continuous parameter space.
>
> Q2: “it would be advisable to provide additional experimental results with a consistent number of training steps for all algorithms to ensure a fair comparison.”
>
> A2: Thanks for your suggestion, before we only show the result when DLPA has already converged in Figure 3. We aim to use Figure 3 to show the superior sample efficiency achieved by DLPA. Table 1 shows the asymptotic performance where all the other algorithms have fully converged. Even after giving all the other baselines a significantly larger number of samples, DLPA still achieves better performance in 6 out of the 8 tested domains and comparable performance in the other 2. **We have also updated the plots for the comparison of all methods with full steps in Figure 6 in the appendix**, where the results for the baselines are very close to what Li et al. 2022 reported in their paper. Note that all the experiments were run on a Nvidia GeForce RTX 3090 GPU. We directly use the source code provided by the authors for those baselines and so the resources are kept the same for all benchmarks. We also run DLPA for an additional amount of steps after convergence for each environment, and the performance is still quite stable as shown in Figure 6.

---

> ### Author Response · Authors · 2023-11-20
> **Official Comment by Authors (Part 2)**
>
> Q3: “For the experiments assessing the impact of CEM-based MPC on DLPA, it would be beneficial to include ablation studies with other traditional model-based non-MPC methods, such as MBPO, applied in the context of PAMDP.”
>
> A3: Thanks for your suggestion. We add the comparison to the Dyna-like model-based RL method in Section 6.3, which is a baseline very close to MBPO that does not use MPC. We use a model-free RL framework for PAMDP—PDQN—as the policy learning framework given the data generated from the learned model. We have also tuned the proportion between the generated data and the real-world rollouts to make sure the baseline reaches its best performance. As we can see from the training results on two benchmarks,  this non-MPC baseline generally performs better than the model-free baselines but is not as good as DLPA.
>
> Q4: “What specific form do the parameters of CEM take in this context? Are they similar to the parameters of a policy neural network, and how do they relate to the state input?”
>
> A4: The parameters of CEM in DLPA directly control the distribution over the discrete actions and continuous parameters and are different from the parameters for a policy neural network. Specifically, for discrete actions, the parameters of CEM are the parameters for a multinomial distribution. We sample from this distribution to generate discrete action during planning. For continuous parameters, the parameters of CEM are the parameters for a Gaussian distribution. We directly sample from it to generate the corresponding continuous parameters during planning. After randomly initializing the action distributions, we use the learned model and the **initial state input** to rollout imagination trajectories and estimate their cumulative return. And we pick the actions to execute based on the estimated returns. Note that for each planning step, we will update the CEM parameters for several iterations (described by Eqn 5, 6, 7) given the estimated returns.
>
> Q5: “How does the computational complexity of CEM changes as the action space expands?”
>
> A5: We have included a new Table in appendix 8.6 that compares the clock time of planning and the number of timesteps needed to converge as the action space expands. We tested this on the Hard Move domain, where the number of discrete actions changes from $2^4$ to $2^{10}$ (one continuous parameter for each of them). As shown in the table, while the number of samples increases as the action space expands, it’s still within an acceptable range even when it’s extremely large. We also quantify this with theoretical analysis which is included as a new section in the updated manuscript.

---

> > ### Comment · Reviewer_75cb · 2023-11-21
> >
> > Thanks for your reply! My concerns have been addressed.

---

> > > ### Author Response · Authors · 2023-11-21
> > >
> > > Thank you! Is there anything else that is holding you back from raising your score that we can address?

---

### Official Review · Reviewer_7BWk · 2023-10-31

**Soundness:** 2 fair
**Presentation:** 2 fair
**Contribution:** 1 poor
**Rating:** 5
**Confidence:** 4

**Summary:**

This paper applies a model-based RL method to the parameterized action MDP (PAMDP).

**Strengths:**

- The paper is easy to understand.

**Weaknesses:**

- Novelty and contribution is limited.

**Questions:**

- What is the research question you study? In other words, what is the specific difficulty or particular problem when applying existing model-based RL methods on PAMDP tasks?

---

> ### Author Response · Authors · 2023-11-20
>
> Q: What is the research question you study? In other words, what is the specific difficulty or particular problem when applying existing model-based RL methods on PAMDP tasks?
>
> A: Thank you for the review. The research question of this paper centers on Reinforcement learning problems where actions are of a mixed type and interactions are readily described by Parameterized Action MDPs (PAMDPs). Here we would like to emphasize again that the primary contribution of this paper is the first model-based RL algorithm for the PAMDP setting, which outperforms all the previous model-free RL methods for PAMDP on 8 standard PAMDP benchmarks. PAMDP is a very common problem setting in real-world tasks, and has gained a lot of attention in recent papers despite all of them being model-free methods.  We hope our paper can draw other researchers' attention to the model-based aspect of this problem setting since DLPA already performs better than the model-free methods without very fancy designs.
>
> In terms of the specific design when applying model-based RL on PAMDP tasks, besides the multi-step prediction technique which is also mentioned by the other reviews, **one major modification we make when applying MPPI to PAMDPs is that we keep a separate distribution over the continuous parameters for each discrete action and update them at each iteration instead of keeping just one independent distribution for the discrete actions and one independent distribution for the continuous parameters.** In other words, we let the distribution of the continuous parameters condition on the chosen discrete action during the sampling process. This is an important change to make when using MPPI for PAMDPs as we don’t want to throw away the established dependency between the continuous parameter and corresponding the discrete action in PAMDPs.  In the updated manuscript we have included a new ablation study result for this change which shows that this change is indispensable for DLPA to do proper planning in PAMDP (Section 6.3).
>
> In terms of novelty, we have also added an entirely new theoretical analysis section in the updated manuscript where we derive some performance guarantees / bounds for the proposed algorithm (Section 5). In general, we provide three main theorems that describe the performance guarantees of DLPA from three aspects: 1. A theoretical bound for the cumulative return difference between the rollout trajectories of DLPA and the optimal trajectories. 2. A theoretical bound for the multi-step prediction error of DLPA. 3. A theoretical bound for how the estimation error will change with respect to the number of samples and the cardinality of the continuous parameter space.

---

> > ### Comment · Reviewer_7BWk · 2023-11-22
> > **Thanks for interpretation**
> >
> > I appreciate the efforts made by the authors. I would raise my score to weak rejection as I still don't think the research question is significant and the contribution of the proposed method is modest.

---

### Official Review · Reviewer_bdkF · 2023-11-01

**Soundness:** 2 fair
**Presentation:** 2 fair
**Contribution:** 2 fair
**Rating:** 5
**Confidence:** 4

**Summary:**

The paper introduces DLPA - a model based RL approach that brings together parametrized action MDPs with model predictive control. The method was applied to 8 environments that previous works have used and demonstrate up to 30x sampling efficiency.

**Strengths:**

The key difference between previous work and DLPA is that model learning is performed by relying just on the initial state and the actions trajectory instead of all intermediate transitions. This provides better inductive bias for longer time horizon tasks and is show to be a critical component by the ablation study in section 5.3.

The reported results show that DLPA learns much faster (in terms of number of samples) compared to previous benchmarks which again can be the result of learning longer horizon models.

**Weaknesses:**

The paper brings together PAMDPs and model predictive control in a very conventional way and so I do not find DLPA novel enough from a technical perspective. In terms of results, despite the reported sampling efficiency, even though the testing environments are used by other previous works, they still seem to be relatively simplistic even compared to other game based benchmarks such as ATARI.

**Questions:**

Also, the reported results both in Figure 3 and Table 1 seem to much lower than the results reported in Li et al. 2022 and their HyAR approach. How much resources did you spend training the benchmarks versus your DLPA? My biggest concern is that the reported sample efficiency is caused by better hyper parameter tuning.

Learning models conditioned just on the initial state and not intermediate transitions must be much more difficult in stochastic environments (either stochastic transitions or actions), have you done any analysis in this direction?

---

> ### Author Response · Authors · 2023-11-20
> **Official Comment by Authors (Part 1)**
>
> Thank you for the thoughtful review. We believe we can address your concerns.
>
> Q1: “The paper brings together PAMDPs and model predictive control in a very conventional way and so I do not find DLPA novel enough from a technical perspective.”
>
> A1: DLPA, while perhaps conventional, is the very first model-based RL algorithm for the PAMDP setting. It outperforms all the previous model-free RL methods - many of which have been published at ICLR, including one just last year - for PAMDP on 8 standard PAMDP benchmarks. The PAMDP setting is very common in real-world tasks, and has drawn a lot of attention in recent papers, but all of them are model-free methods.  We hope our paper can draw other researchers’ attention to the model-based aspect of this problem setting since DLPA already performs better than the model-free methods, even the most complex ones.
>
> Meanwhile, we believe our algorithm also has some unconventional techniques which we found to be useful in practice. For example, we use the initial state and intermediate predictions to do model learning. Moreover, for using MPPI for PAMDPs, previous works have only shown its efficiency on either discrete action space or continuous action space, while we’re the first to apply such a method to the parameterized action space. **The main modification we make when applying MPPI to PAMDPs is that we keep a separate distribution over the continuous parameters for each discrete action and update them at each iteration instead of keeping just one independent distribution for the discrete actions and one independent distribution for the continuous parameters.** In other words, we let the distribution of the continuous parameters condition on the chosen discrete action during the sampling process. We believe this is an important change to make when using MPPI for PAMDPs as we don’t want to throw away the established dependency between the continuous parameter and corresponding discrete action in PAMDPs. **In the updated manuscript we have included new ablation study results for this change which shows that this change is indispensable for DLPA to do proper planning in PAMDP** (Section 6.3).
>
>
> In terms of novelty, we have also added an entirely new theoretical analysis section in the updated manuscript where we derive some performance guarantees / bounds for the proposed algorithm (Section 5). In general, we provide three main theorems that describe the performance guarantees of DLPA from three aspects: 1. A theoretical bound for the cumulative return difference between the rollout trajectories of DLPA and the optimal trajectories. 2. A theoretical bound for the multi-step prediction error of DLPA. 3. A theoretical bound for how the estimation error will change with respect to the number of samples and the cardinality of the continuous parameter space.
>
> Q2: “even though the testing environments are used by other previous works, they still seem to be relatively simplistic even compared to other game based benchmarks such as ATARI.”
>
> A2: We first want to point out that all these 8 testing environments are the ones used in an ICLR paper for PAMDP last year. The PAMDP setting is quite recent and so the leading experimental benchmarks are not as complex as the MDP setting. We agree with the review that evaluating and developing our algorithm on more complex PAMDPs is one of our main future directions. But in order to address the reviewer’s concern, we further test our method on a much more complex domain Half-Field-Offence (HFO) domain [1], where both the state space and action space are much larger. Besides, the task horizon is much longer and there is more randomness existing in the environment. HFO is originally a subtask in RoboCup simulated soccer. We are working on this domain and will update the manuscript with the new results soon. **Update: we have added the new results in appendix Section 8.7. DLPA still outperforms the other model-free PAMDP baselines in this much more complicated domain.**
>
> [1] Hausknecht, M., Mupparaju, P., Subramanian, S., Kalyanakrishnan, S., & Stone, P. (2016, May). Half field offense: An environment for multiagent learning and ad hoc teamwork. In AAMAS Adaptive Learning Agents (ALA) Workshop (Vol. 3). sn.

---

> ### Author Response · Authors · 2023-11-20
> **Official Comment by Authors (Part 2)**
>
> Q3: “the reported results both in Figure 3 and Table 1 seem to much lower than the results reported in Li et al. 2022 and their HyAR approach. How much resources did you spend training the benchmarks versus your DLPA?”
>
> A3: This is because, in Figure 3, we limit the x-axis, the time steps, to the point where our DLPA has already converged. We aim to use Figure 3 to show the superior sample efficiency achieved by DLPA. Table 1 shows the asymptotic performance where all the other algorithms have fully converged. Even after giving all the other baselines a significantly larger number of samples, DLPA still achieves better performance in 6 out of the 8 tested domains and comparable performance in the other 2. We also show the plots for the comparison of all methods with full steps in Figure 6 in the appendix, where the results for the baselines are very close to what Li et al. 2022 reported in their paper. Note that all the experiments were run on a Nvidia GeForce RTX 3090 GPU. We directly use the source code provided by the authors for those baselines and so the resources are kept the same for all benchmarks.
>
> Q4: “Learning models conditioned just on the initial state and not intermediate transitions must be much more difficult in stochastic environments (either stochastic transitions or actions), have you done any analysis in this direction?”
>
> A4: Thanks for pointing this out. Actually all the 8 domains we tested are stochastic in terms of the transition function. And our ablation studies have shown the superiority of this approach in the tested environments. We have included one more ablation study results plot on the Platform environment in the updated manuscript. As the reviewer mentioned, this approach can be much more difficult in stochastic environments, so we multiply a discount factor along the multi-step prediction in Eqn. (4) to help stabilize training.

---

### Official Review · Reviewer_WRCa · 2023-11-03

**Soundness:** 4 excellent
**Presentation:** 4 excellent
**Contribution:** 3 good
**Rating:** 10
**Confidence:** 4

**Summary:**

This paper considers reinforcement learning problems in settings where actions are of a mixed type and interactions are readily described by Parameterized Action MDPs (PAMDPs). In this setting, the paper asks whether there exist model-based methods that can efficiently find solutions.

The paper proposes a model-predictive control algorithm called DLPA, and it thoroughly evaluates it against current algorithms on PAMDP baselines.

The paper makes the following contributions.
 * A model-based RL method for PAMDPs called Dynamics Learning and predictive control with Parameterized Actions (DLPA).
 * DLPA is the first model-based algorithm for the PAMDP setting.
 * Empirical evidence that DLPA performs effectively in PAMDP benchmarks.

I really enjoyed reading this paper. Aside from some minor issues, I think the study was well executed and makes clear contribution. I elaborate on my position below.

**Strengths:**

* The paper is generally well written. I was able to easily identify the research questions and understand the main contributions.
* The empirical study is excellent. The experiments are constructed around the standard methodology which is aimed at supporting its main claim for SOTA performance in PAMDP benchmarks. The benchmarks and baselines seem reasonable and fairly chosen. The results provide positive evidence for its SOTA claim and demonstrate a significant performance margin between the proposed method and other baselines. The experiment in Section 5.3 addresses a nuanced claim made in Section 4.1---supporting that with clear positive evidence.
* The paper makes a significant contribution.

**Weaknesses:**

* Many of my comments on this paper are minor.
* The proposed algorithm seems limited to small-horizon problems, as its backward-pass computation and the number of parameters both seem to scale linearly in the horizon length.
* The proposed algorithm has a lot of hyperparameters, which could make the algorithm difficult to tune.

**Questions:**

General comments and questions
 * Section 2.2: "model to take in ~~this~~ actions"
 * Section 2.2: From what state are the action sequences sampled?
 * Section 2.2: It would be helpful to include a reference to introductory material on Model Predictive Control.
 * Section 3: This dichotomy of model-based methods confused me, because the two categories (random shooting and data augmentation) don't entire split the class of methods. Here, my issue is with random shooting. I suggest using a more descriptive and potentially conventional dichotomy which splits the class based on how the model is used in planning, such as Dyna-based methods and decision-time planning methods. One uses planning for credit assignment and the other for policy evaluation.
 * Section 4: Section 4.1 and Figure 1 point to notation that isn't defined until Section 4.2. Consider introducing notation earlier.
 * Figure 1: The illustrations are nicely done here. However, the caption could use more supporting text so readers understand the semantics of the green dots and the trajectories inside the states.
 * Section 4.1: Why do you call $s_t$ an observation when it was previously defined as a state in Section 2 (assuming it comes from $S$ )? If you want to distinguish it from outputs of the dynamics model, then you can always use "environment state" and "model state."
 * Section 4.1: Should $r_{t+1}$ have a hat, since it is an estimated quantity?
 * Section 4.1: "by learning to inferring" --> "by learning to infer".
 * Section 4.1: The proposed loss computes errors between the observed transition quantities and the respective predictions at each step---the latter of which are, importantly, functions of intermediate predictions. The authors claim this choice is preferable to using observed quantities in place of the intermediate predictions. Presumably, this choice allows gradients to flow back through time and assign credit more effectively than the alternative. The last few sentences of the final paragraph could make this point more clearly.
 * Section 4.2: Defining the parameters before they are used in $\mathcal{C}^0$ would add clarity to this section.
 * Section 4.2: Equation 4 should explicitly state that $\hat{s}_{t_0}=s_{t_0}$ .
 * Section 4.2: Are your action distributions are unconditional on any context?
 * Section 4.2: You are overloading notation for $d$ . Consider using $\text{d}$ or some other notation do avoid overloading.
 * Algorithm 1: "Execute the first action ..."---It would add clarity to write out the action using your notation.
 * Algorithm 1: The "training" variable is never defined, so it is unclear why the conditional statement is included. I understand what the paper hopes to communicate here: there is an initial period of data gathering with no updates. However, the pseudocode should either reflect this more accurately or it should exclude this logic---considering it as an implementation detail.
 * Figure 3: It would be more accurate to label the vertical axis "Average Episodic Return", as the reward is just a momentary quantity.
 * Table 1: I need help understanding these results, because they don't seem consistent with Figure 3. In Figure 3., DLPA achieves the highest performance in Platform and Goal. Why then is HyAR bolded?
 * Section 5.2: "action-spercialized"?
 * Section 5.3: "that just do random shooting"---fix grammar.
 * Section 5.3: The last paragraph needs a grammatical revision.
 * Section 5.3: The last paragraph could do a better job explaining why Random Shooting fails.
 * This algorithm seems to use a heavy amount of computation. Can you comment on the algorithm's complexity?

---

> ### Author Response · Authors · 2023-11-20
>
> Thanks for your very positive comments and thoughtful suggestions! We correct the grammar and presentation issues according to your kind advice and address your concerns below.
>
> Q1: From what state are the action sequences sampled?
>
> A1: Action sequences are sampled starting from the current timestep t (so the initial state s_t) until timestep t+H from randomly initialized distributions like Gaussians.
>
> Q2: Suggestions for references, notations and other writing issues.
>
> A2: Thanks for pointing all these out! We have updated the manuscript accordingly.
>
>
> Q3: “Are your action distributions unconditional on any context?”
>
> A3: At the first timestep of each iteration for planning, we just set the distribution over discrete actions to be a multinomial distribution with parameter \theta_k^0 = 1/K, and set the distributions over the corresponding continuous parameters to be a multivariate Gaussian distribution with parameters \mu_k^0 = 0, \sigma_k^0 = 1. Then for all the timesteps afterward, the action distributions are conditioned on the distributions at the last timestep—weighted by the cumulative return.
>
>
> Q4: “I need help understanding these results, because they don't seem consistent with Figure 3. In Figure 3., DLPA achieves the highest performance in Platform and Goal. Why then is HyAR bolded?”
>
> A4: This is because, in Figure 3, we limit the x-axis (time steps) to the point where our DLPA has already converged. We aim to use Figure 3 to show the superior sample efficiency achieved by DLPA. Table 1 shows the asymptotic performance where all the other algorithms have fully converged. Even after giving all the other baselines a significantly larger number of samples, DLPA still achieves better performance for 6 out of the 8 tested domains and comparable performance on the other 2.
>
>
> Q5: “This algorithm seems to use a heavy amount of computation. Can you comment on the algorithm's complexity?”
>
> A5: We have included a new Table in appendix 8.6 that compares the clock time of planning and the number of timesteps needed to converge as the action space expands. We tested this on the Hard Move domain, where the number of discrete actions changes from $2^4$ to $2^{10}$ (one continuous parameter for each of them). As shown in the table, while the number of samples increases as the action space expands, it’s still within an acceptable range even when it’s extremely large. We also quantify this with theoretical analysis which is included as a new section in the updated manuscript.

---

### Official Review · Reviewer_RWZZ · 2023-11-06

**Soundness:** 3 good
**Presentation:** 3 good
**Contribution:** 2 fair
**Rating:** 6
**Confidence:** 4

**Summary:**

This paper addresses the problem of model-based reinforcement learning for parameterized action markov decision processes (PAMDPs), by proposing a framework referred to as Dynamics Learning and Predictive Control with Parameterized Actions (DLPA). The core idea of DLPA lies in the integration of neural-network based model learning (transition model, termination model, and reward function) and the model predictive control using model predictive path integral. Extensive experiments against multiple baseline algorithms for PAMDP are provided to demonstrate the efficiency of DLPA.

**Strengths:**

+ The proposed framework is technically sound and the key idea of DLPA is clear.

+ Extensive experiments are given against multiple baseline algorithms.

**Weaknesses:**

- The novelty seems rather limited as the main framework of DLPA follows the standard data-driven MPPI, with the contextualization from MDP to PAMDPs. For example, an information theoretic MPC framework was proposed in [G. Williams et al, ICRA 20217] to incorporate the model learning in the MPPI-based planning procedure.

- Some key information is not provided, e.g. which neural network algorithms are used to optimize Eq. 1?

- No theoretical analysis is given to justify the performance applied to PAMDPs, which seems necessary to help readers understand the significance of the DLPA beyond simple contextualization of data-driven MPPI to PAMDPs. As PAMDPs concerned about finding the right action and the associated parameters to use, is there any guarantee of convergence to the selected parameterized actions during training from DLPA?

**Questions:**

1. What is the main contribution behind DLPA, or could authors comment on specific challenges solved when using MPPI for PAMDPs?

2. What algorithms are used to solve Eq. 1 in terms of model learning? Also, it is unclear why the authors imply that the ground truth state was not used as input to train their model in DLPA. For example, the ground-truth state s_{t+1} is needed to compute the loss L_joint defined in Eq. 1.

3. Does DLPA deliver any theoretical properties to justify its performance on PAMDPs? Authors are encouraged to discuss the convergence of parameterized actions from the DLPA framework, if any.

---

> ### Author Response · Authors · 2023-11-20
>
> Thank you for the thoughtful review We address your concern as follows:
>
> Q1: “What is the main contribution behind DLPA, or could authors comment on specific challenges solved when using MPPI for PAMDPs?”
>
> A1: The primary contribution is the first model-based RL algorithm for the PAMDP setting, which outperforms all the previous model-free RL methods for PAMDP on 8 standard PAMDP benchmarks. The PAMDP model is appropriate to many very common real-world tasks, and has gained a lot of attention in recent papers - however, all existing methods are model-free.  We hope our paper can draw other researchers' attention to the model-based aspect of this problem setting since DLPA performs better than even the most sophisticated model-free methods. In terms of novelty, inspired by the reviewer’s question, **we have added a new theoretical analysis section in the updated manuscript where we derive some performance guarantees / bounds for the proposed algorithm.** Thank you for the suggestion!
>
> For using MPPI for PAMDPs, previous works have only shown its efficiency on either discrete action space or continuous action space, while we’re the first to apply such a method to the parameterized action space. **The main modification we make when applying MPPI to PAMDPs is that we keep a separate distribution over the continuous parameters for each discrete action and update them at each iteration instead of keeping just one independent distribution for the discrete actions and one independent distribution for the continuous parameters.** In other words, we let the distribution of the continuous parameters condition on the chosen discrete action during the sampling process. This is an important change to make when using MPPI for PAMDPs as we don’t want to throw away the established dependency between the continuous parameter and corresponding the discrete action in PAMDPs.  In the updated manuscript we have included a new ablation study result for this change which shows that this change is indispensable for DLPA to do proper planning in PAMDP (Section 6.3).
>
>
> Q2: “What algorithms are used to solve Eq. 1 in terms of model learning? Also, it is unclear why the authors imply that the ground truth state was not used as input to train their model in DLPA. For example, the ground-truth state s_{t+1} is needed to compute the loss L_joint defined in Eq. 1.”
>
> A2: To solve Eqn. (1), we simply learn three neural networks for the reward predictor, transition predictor and termination predictor respectively using supervised learning. And we use the mean square error as the loss of the state and reward prediction, and use cross entropy loss as the objective function for updating the termination signal. We train our model based on multi-step prediction (only the ground truth state at step 1 is needed) to encourage the model to do better H-step long-term prediction. This design for the training process is consistent with the planning process, as H is also the planning horizon in MPPI where we do the same long-term prediction to select actions. As also pointed out by reviewer WRCa, this choice allows gradients to flow back through time and assign credit more effectively than the alternative, which is one-step training. We found this design choice to be especially useful during empirical evaluation as shown in Section 6.3.  We have also included one additional experimental results plot on Platform to show this in the updated manuscript.
>
> Q3: “Does DLPA deliver any theoretical properties to justify its performance on PAMDPs? Authors are encouraged to discuss the convergence of parameterized actions from the DLPA framework, if any.”
>
> A3: We thank the reviewer for pointing this out. We have added a theoretical analysis section (also in the appendix) in the updated manuscript (section 5). In general, we provide three main theorems that describe the performance guarantees of DLPA from three aspects: 1. A theoretical bound for the cumulative return difference between the rollout trajectories of DLPA and the optimal trajectories. 2. A theoretical bound for the multi-step prediction error of DLPA. 3. A theoretical bound for how the estimation error will change with respect to the number of samples and the cardinality of the continuous parameter space.

---

> > ### Comment · Reviewer_RWZZ · 2023-11-23
> > **Thanks for the additional information**
> >
> > Thanks for the additional information regarding the theoretic analysis and response to my questions. I agree the contextulization of MPPI to the PAMDP will be useful to the community, although the technical novelty may not be that significant. Given the authors' responses, I am happy to raise my score by one level up to "marginally above the acceptance threshold".

---

### Author Response · Authors · 2023-11-20
**New Results and Updated Manuscript**

In response to the reviews, we have updated our manuscript and we list the major modifications below:

1. We have included a new theoretical analysis in section 5 and proofs & additional theorems in the appendix. In general, we provide three main theorems that describe the performance guarantees of DLPA from three aspects: a. A theoretical bound for the cumulative return difference between the rollout trajectories of DLPA and the optimal trajectories. b. A theoretical bound for the multi-step prediction error of DLPA. c. A theoretical bound for how the estimation error will change with respect to the number of samples and the cardinality of the continuous parameter space.

2. Additional ablation studies on non-MPC model-based RL methods, computational complexity, multi-step prediction during model learning, and regular MPPI.

3. We are running on an additional domain “Half Field Offense” (HFO) and will update the results soon. **Update: we have added the new results in appendix Section 8.7.**

4. We have fixed the writing issues mentioned by the reviewers.

Regarding the novelty of our paper, we acknowledge the reviewers' concerns and hope our additional theoretical analysis and experimental results have elucidated the unique aspects of our work. We aspire to contribute meaningfully to the field and believe that the **simplicity** of our new algorithm, coupled with its ability to **outperform all the other existing PAMDP RL algorithms**, represents a significant advancement.

---

### Comment · Area_Chair_zSDH · 2023-11-22

Dear reviewers,

This a reminder that deadline of author/reviewer discussion is AOE Nov 22nd (today). Please engage in the discussion, check if your concerns are addressed, and make potential adjustments to the rating and reviews.

Thank you!
AC

---

### Meta-Review · Area_Chair_zSDH · 2023-12-06

**Metareview:**

This work proposes a model-based RL method to solve Parameterized Action MDP. The proposed method, Dynamics Learning and predictive control with Parameterized Actions (DLPA), learns a model that conditions on the initial state and the predicted action trajectory, and then adopts model predictive control for planning. The continuous parameters are conditioned on the chosen discrete action during sampling, which is a key difference to MPPI. The reviews recognize that the paper is well-written despite quite a few grammar errors, and that the experiments show good performance especially for sample efficiency. The authors made significant revisions to the paper to address the reviewers' questions. Some weaknesses remain, including limitations to small-horizon problems (WRCa), difficulty of tuning the algorithm's hyperparameters (WRCa), and lack of novelty and contribution (75cb, 7BWk, bdkF, RWZZ). While the authors added new theoretical analyses, it is unclear how these analyses on bounds guarantee high performance of DLPA, and why the bounds are significant or useful. During the discussion phase, reviewers (including Reviewer RWZZ and Reviewer 7BWk) again expressed concerns on the limited novelty and significance of the technical contribution, given that the proposed method only does minor modification to MPPI. I suggest the authors carefully consider the feedback from all reviewers and potentially improve the clarity on novelty and contribution. Unfortunately, given the shared concerns, I recommend to reject this paper.

**Justification For Why Not Higher Score:**

Some weaknesses remain, including limitations to small-horizon problems (WRCa), difficulty of tuning the algorithm's hyperparameters (WRCa), and lack of novelty and contribution (75cb, 7BWk, bdkF, RWZZ). While the authors added new theoretical analyses, it is unclear how these analyses on bounds guarantee high performance of DLPA, and why the bounds are significant or useful. During the discussion phase, reviewers (including Reviewer RWZZ and Reviewer 7BWk) again expressed concerns on the limited novelty and significance of the technical contribution, given that the proposed method only does minor modification to MPPI.

**Justification For Why Not Lower Score:**

n/a

---

### Decision · Program_Chairs · 2024-01-16

Reject